# Multi-Scale Generative Modeling in Wavelet Domain

## Abstract

While working within the spatial domain can pose problems associated with ill-conditioned scores, recent advancements in diffusion-based generative models have shown that transitioning to the wavelet domain offers a promising alternative. However, within the wavelet domain, we encounter unique complexities, especially the sparse representation of high-frequency coefficients, which deviates significantly from the Gaussian assumptions in the diffusion process. To this end, we propose developing a multi-scale generative model directly within the wavelet domain using Generative Adversarial Networks. This Multi-Scale Generative Model in the Wavelet Domain (i.e., Wavelet Multi-Scale Generative Model (WMGM)) leverages the benefits of wavelet coefficients, with a specific emphasis on using low-frequency coefficients as conditioning variables. Based on theoretical analysis and experimental results, our model provides a pioneering framework for implementing generative models in the wavelet domain, showcasing remarkable performance improvements and significant reduction in trainable parameters, sampling steps and time. This innovative approach represents a promising step forward in the field of diffusion modeling techniques.

## 1 Introduction

Generative model (GM) aims to extract knowledge and data from noise. Simply speaking, the GM takes as input a large amount of data and uses a loss function that encodes the notion of likeness (i.e., measuring the differences between the empirically observed and the generated distributions) in order to construct/generate a high-dimensional probability distribution that approximates the true data distribution. With this goal in mind, over the last decades, several GMs have been proposed, such as reinforcement learning (RL), variational autoencoders (VAE), recurrent neural networks (RNNs), generative adversarial networks (GANs), or hybrid (i.e., combine the previously mentioned strategies) approaches, providing strategies for generating high-quality audio waveforms or speech (Oord et al. (2016)), constructing natural-looking images (Goodfellow et al. (2014); Brock et al. (2018)), generating coherent text (Bowman et al. (2015)), designing molecules (Gómez-Bombarelli et al. (2018)). Within the GM domain, score-based generative models (SGMs) (Song et al. (2020); Ho et al. (2020); Song & Ermon (2019)), also known as denoising diffusion models, construct and encode the probability distributions through a scoring approach (e.g., a vector field that points in the direction of increasing likelihood of data) and recover the actual data distribution through a learnable reverse process to the forward Gaussian diffusion process. Several works have successfully shown that these score functions can be learned from data without requiring adversarial optimization and producing realistic samples.

Although these SGMs have been highly successful, many physical, chemical, biological, and engineering systems display complex multiscale and non-Gaussian properties, leading to ill-conditioned scores. To deal with non-Gaussian high-dimensional probability distribution functions (HDPDFs), Guth et al. (2022b) proposed to factorize these HDPDFs into products of conditional probabilities of normalized wavelet coefficients across scales and introduced the wavelet score-based generative model (WSGM). This WSGM constructs the conditional probabilities of normalized wavelet coefficients exploiting the same discretization schedule across all scales. WSGM offers insightful inspiration regarding the feasibility of conducting generative operations within the wavelet domain through wavelet coefficients.

To make the discussion more concrete, let us consider the noise-adding process in the frequency domain (i.e., wavelet domain), where noise contains a uniform power spectrum in each frequency

band. However, in the wavelet domain, the high-frequency coefficients of the images are sparse and contain minimal energy, while the low-frequency coefficients encapsulate most of the energy. Given the disparity between image and noise power spectra, low-frequency components, which hold the majority of the energy, receive the same magnitude of noise during the noise addition process, and high-frequency coefficients, despite being sparse, obtain a relatively larger amount of noise. This dynamic offers inspiration for analyzing diffusion in the frequency domain, incorporating corresponding generative strategies tailored for each frequency sub-domain (namely, the high and low-frequency domains). Endeavors have also been undertaken to do the diffusion in the frequency domain. Some studies highlight the challenges associated with the ill-conditioning properties of the score during diffusion in the spatial domain. They astutely establish a connection between the minimum discretization steps and the power spectrum of the image, thereby rationalizing the application of noise addition in the wavelet domain.

While the advantages of conducting diffusion within the wavelet domain are evident from both experimental results and intuitive understanding, there is a lack of theoretical substantiation for the correctness of general diffusion in the wavelet domain and comprehensive analysis of the properties of each wavelet subband. Specifically, considering the sparsity and the non-Gaussian HDPDF of high-frequency coefficients, discussion about adapting SGM to high-frequency subbands is insufficient. This void in theoretical grounding and sparsity consideration impedes many works from efficiently conducting diffusion in the wavelet domain Huang et al. (2023); Jiang et al. (2023). To bridge this gap, our research first highlights that deep-scale low-frequency coefficient scores are well-conditioned, alleviating the ill-conditioning issue by general diffusion models in the spatial domain. Subsequently, we illustrate that there is a duality between SGM in the wavelet domain and the spatial domain. Furthermore, considering high-frequency coefficients' sparsity and non-Gaussian nature, we introduce Generative Adversarial Neural Operators for multi-scale generative operator learning. The framework facilitates high-quality generation, data efficiency, and fast sampling.

**Our contributions**: (*i*) We are the first to explicitly demonstrate the duality between the spatial and the wavelet domain in terms of diffusion and reverse processes; (*ii*) we theoretically establish the SGM framework in the wavelet domain and analyze the distribution of multi-scale wavelet coefficients; (*iii*) we introduce the multi-scale generative adversarial neural operator (GANO) in the wavelet domain to tackle the highly non-Gaussian distribution of high-frequency wavelet coefficients efficiently; (*iv*) we demonstrate that the proposed WMGM framework, as depicted in Fig. 1 has a faster sampling speed and better generation quality than competitive methods.

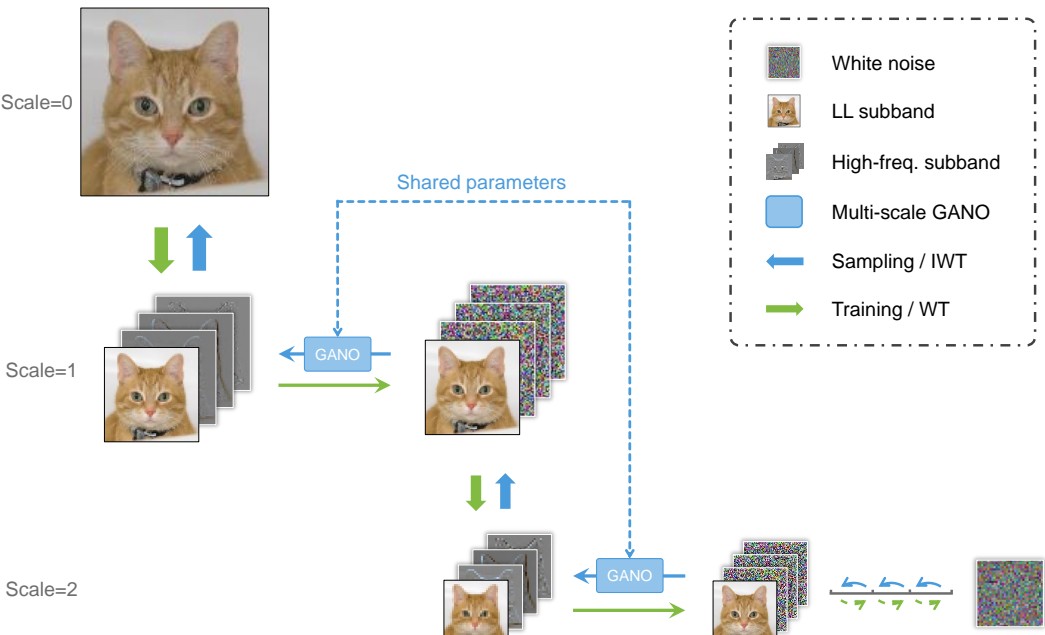

Figure 1: The framework of the proposed WMGM. WT, IWT: wavelet transform and inverse wavelet transform.

## 2    RELATED WORK

Diffusion models have emerged as state-of-the-art generative models, which are stable and capable to generate high fidelity images. They leverage the Markov chains to gradually introduce noises into the data and learn to capture the reverse process for sampling. To optimize the reverse process, Denoising diffusion probabilistic models (DDPMs) Ho et al. (2020); Dhariwal & Nichol (2021) predict the added noises from the diffused output at arbitrary time steps. Another common approach is score-based generative models (SGMs) Song & Ermon (2019); Song et al. (2020), which aims to predict the score function $\nabla \log(p_{x_t})$. Also, a recent study named 'Cold Diffusion' explores the possibility of using deterministic transformations instead of random Gaussian noise in diffusion models Bansal et al. (2022). Another major disadvantage of diffusion models remains that they take long-time to sampling. To this end, some recent studies have applied to accelerate the sampling process Xiao et al. (2021); Wang et al. (2022).

Emerging works attempt to incorporate the wavelet domain into diffusion models, either to facilitate the training or to speed up the sampling process. WSGM Guth et al. (2022b) proposes multi-scale score-based diffusion model and theoretically analyzes the advantages of wavelet-domain representation. WaveDiff Phung et al. (2023) applies GAN to accelerate the prediction of wavelet coefficients during the reverse process in wavelet domain. Huang et al. (2023) leverages low-frequency information from the first 3 bands for the diffusion process, and proposes an efficient sampling strategy conditioned on degraded images and predicted high spectrum for image restoration. The effectiveness of adaptation of diffusion to wavelet domain has also been successfully demonstrated for a wide variety of tasks, including low-light image enhancement Jiang et al. (2023), image super-resolution Moser et al. (2023) and 3D shape generation Hui et al. (2022).

## 3    GENERATIVE MODELING IN THE WAVELET DOMAIN

In the introduction, we discuss the intuitive limitations of diffusion in the spatial domain, as also depicted in Fig. 2, where noise is added in the spatial domain gradually and the resulting noise figures are transformed into the wavelet domain. In this section, we mathematically elucidate the problems of conducting diffusion in the spatial domain from the perspective of the ill-conditioning score. Furthermore, we propose conducting diffusion processes in the wavelet domain as a solution to address these issues.

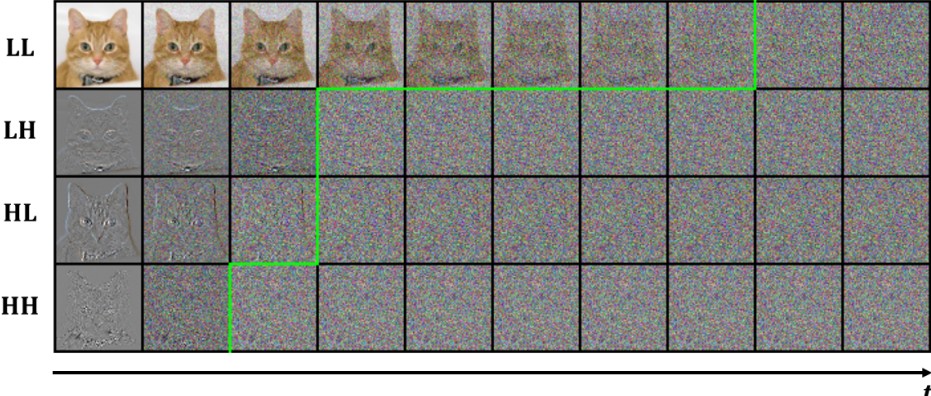

Figure 2: Diffusion trajectories of the wavelet coefficients. Notice that the high-frequency components (LH,HL,HH) are overwhelmed by noise at an earlier stage marked by the green line. At the same time, the low-frequency component (LL) degrades more slowly.

### 3.1    ILL-CONDITIONIONG SCORE IN THE SPATIAL DOMAIN

For the Score-based Generative Model (SGM) Song et al. (2020); Ho et al. (2020); Song & Ermon (2019), the forward/noising process can be mathematically formulated as the Ornstein-Uhlenbeck (OU) process. The general time-rescaled OU process can be written as follows:

$$dX_t = -g(t)^2 X_t dt + \sqrt{2}g(t)dB_t. \tag{1}$$

Here, $(\boldsymbol{X}_t)_{t\in[0,T]}$ is the noising process with the initial condition $\boldsymbol{X}_0$ sampled from the data distribution. $(\boldsymbol{B}_t)_{t\in[0,T]}$ is a standard d-dimensional Brownian motion. We use $\boldsymbol{X}_t^{\leftarrow}$ to denote the reverse process, such that $(\boldsymbol{X}_t^{\leftarrow})_{t\in[0,T]} = (\boldsymbol{X}_{T-t})_{t\in[0,T]}$. With the common assumption that $g(t) = 1$ in standard diffusion models, the reverse processes as follows:

$$d\boldsymbol{X}_t^{\leftarrow} = (\boldsymbol{X}_t^{\leftarrow} + 2\nabla \log p_{T-t}(\boldsymbol{X}_t^{\leftarrow})) \, dt + \sqrt{2}d\boldsymbol{B}_t, \tag{2}$$

where $p_t$ denotes the marginal density of $\boldsymbol{X}_t$, and $\nabla \log p_t$ is called the score. To generate $\boldsymbol{X}_0$ from $\boldsymbol{X}_T$ via the time-reversed SDE, it is essential to accurately estimate the score $\nabla \log p_t$ at each time $t$ and to discretize the SDE with minimal error introduction.

An approximation of the reverse process, as given in Eq. 2, can be computed by discretizing time and approximating $\nabla \log p_t$ by $s_t$. This results in a Markov chain approximation of the time-reversed Stochastic Differential Equation (SDE). Accordingly, the Markov chain is initialized by $\tilde{\mathbf{x}}_T \sim \mathcal{N}(0, I_d)$ and evolves over uniform time steps $\Delta t$ that decrease from $t_N = T$ to $t_0 = 0$. The discretized process is as follows:

$$\tilde{\boldsymbol{x}}_{t-1} = \tilde{\boldsymbol{x}}_t + \Delta t \left(\tilde{\boldsymbol{x}}_t + 2s_t(\tilde{\boldsymbol{x}}_t)\right) + \sqrt{2\Delta t}\mathbf{z}_t, \tag{3}$$

where $\mathbf{z}_t$ are realization of Brownian motion $\boldsymbol{B}_t$. Here, the minimum number of time steps $N = \frac{T}{\Delta t}$ is limited by the Lipschitz regularity of the score $\nabla \log p_t$, as detailed in Theorem 1 De Bortoli et al. (2021) and Theorem 3 Guth et al. (2022a). Specifically, let us assume that the data distribution is Gaussian, $p = \mathcal{N}(0, \Sigma)$, with a covariance matrix $\Sigma$ in a $d$-dimensional space. Consider $\tilde{p}_t$ as the distribution corresponding to $\tilde{\mathbf{x}}_t$, the approximation error between the data distribution $p$ and $\tilde{p}_0$ stems from (i) $\Psi_T$: the misalignment between the distributions of $\tilde{\mathbf{x}}_T$ and $\mathbf{x}_t$; (ii) $\Psi_{\Delta t}$: time discretization error. For a detailed expression, see Equations 59. The sum of two errors $\varepsilon$ is connected to $\Sigma$, specifically in the case of uniform time sampling with intervals $\Delta t$. We standardize the signal energy by enforcing $\text{Tr}(\Sigma) = d$ and define $\kappa$ as the condition number of $\Sigma$, which is the ratio of its largest to smallest eigenvalues. Subsequently, the relationship can be expressed as:

$$N = \frac{T}{\Delta t} = o(\epsilon^{-2}\kappa^3) \tag{4}$$

Equation 4 defines the upper bound on the number of time steps $N$. It indicates that as the condition number of the covariance increases, the number of time steps also increases correspondingly. Within the research conducted in Guth et al. (2022a), Theorem 1 is expanded upon to accommodate non-Gaussian processes. This expansion establishes a link between the number of discretization steps $N$ and the regularity of the score $\nabla \log p_t$, as detailed in Theorem 3, suggesting that the non-Gaussian processes characterized by an ill-conditioned covariance matrix necessitate a significant number of discretization steps to achieve a small error.

Natural images are generally modeled by spatially stationary distributions, implying the applicability of Wiener-Khinchin theorem to its covariance matrix Champeney (1987). To be specfic, its covariance matrix $\Sigma$ can be diagonalized by

$$\Sigma = \mathbb{E}(\boldsymbol{X}\boldsymbol{X}^T) = U^T S U. \tag{5}$$

Here $U$ is the matrix represents Fourier transform and $S = diag(S(\omega))$ is the diagonal of the power spectrum of $X$. Therefore, the condition number of such distribution equals to $\frac{\max_\omega(S(\omega))}{\min_\omega(S(\omega))}$. However, according to the well-known power-decay law of the power spectrum of natural images Simoncelli & Olshausen (2001); Ruderman & Bialek (1994), we have

$$S(\omega) \sim (\lambda^\eta + |\omega|^\eta)^{-1}. \tag{6}$$

Natural images generally conform to this power-law decay with $\eta = 1$ and $\frac{2\pi}{\lambda}$ approximating the image width ($L$). Consequently, the reverse process in the spatial domain for natural images is highly ill-conditioned and requires excessive discretization steps to achieve a small error. In order to mitigate the inherent ill-conditioning properties of the score observed in the spatial domain, employing a wavelet transform constitutes a pragmatic solution to this dilemma.

## 3.2 DIFFUSION IN THE WAVELET DOMAIN

In this section, we demonstrate the duality between the spatial and wavelet domains in terms of diffusion and reverse processes to ensure theoretical correctness when performing diffusion within the wavelet domain. Subsequently, we delve into the well-conditioning properties inherent to the low-frequency coefficients and the sparsity of the high-frequency coefficients, serving as a pivotal inspiration for our proposed model. Technical details on wavelet transform and multiresolution analysis (MRA) have been provided in A.7.

### 3.2.1 DUALITY BETWEEN THE SPATIAL DOMAIN AND THE WAVELET DOMAIN

The discrete wavelet transform (DWT), such as Haar DWT (Please refer to A.7.3 for detailed information), facilitates orthogonal transformations within the real domain, and we can smoothly write the DWT as:

$$\hat{\boldsymbol{X}} = \boldsymbol{A}\boldsymbol{X}, \quad \boldsymbol{X} \in \mathbb{R}^d. \tag{7}$$

In this expression, $\hat{\boldsymbol{X}}$ represents the wavelet coefficients of the signal $\boldsymbol{X}$ in $\mathbb{R}^d$, and $\boldsymbol{A}$ denotes the discrete wavelet matrix. Importantly, $\boldsymbol{A}$ is an orthogonal matrix, satisfying the condition $\boldsymbol{A}\boldsymbol{A}^\top = \boldsymbol{I}$, where $\boldsymbol{I}$ is the identity matrix. Following this transformation, we introduce the processes for score-based generative modeling in the wavelet domain.

**Forward process.** We first consider the forward/noising process for the score-based generative modeling process. According to the OU process in the spatial domain formulated in 1, we perform DWT to $\boldsymbol{X}_t$ and figure out $\hat{\boldsymbol{X}}_t$ also observes the similar OU process:

$$d\hat{\boldsymbol{X}}_t = -g(t)^2\hat{\boldsymbol{X}}_t dt + \sqrt{2}g(t)d\hat{\boldsymbol{B}}_t, \quad \hat{\boldsymbol{X}}_0 = \boldsymbol{A}\boldsymbol{X}_0. \tag{8}$$

Here $\hat{\boldsymbol{B}}_t = \boldsymbol{A}\boldsymbol{B}_t$, it is evident that $\hat{\boldsymbol{B}}_t$ is also a standard Brownian motion.

**Reverse process.** We use $\hat{\boldsymbol{X}}_t^{\leftarrow}$ to denote the reverse process in the wavelet domain. According to the reverse process in the spatial domain formulated as Eq. 2, the corresponding process in the wavelet domain is outlined below:

$$d\hat{\boldsymbol{X}}_t^{\leftarrow} = \left( \hat{\boldsymbol{X}}_t^{\leftarrow} + 2\nabla \log q_{T-t}(\hat{\boldsymbol{X}}_t^{\leftarrow}) \right) dt + \sqrt{2}d\hat{\boldsymbol{B}}_t. \tag{9}$$

Here $q_t$ is the density distribution of $\hat{\boldsymbol{X}}_t$, and we have $q_t(\boldsymbol{x}) = p_t(\boldsymbol{A}^T\boldsymbol{x})$. The derivation of the reverse process in the wavelet domain is provided in Appendix A.3. In the training processes, the optimal score-based model $\boldsymbol{r}_{\hat{\theta}}$ approximates the score $\nabla \log q_t$ in the wavelet domain:

$$\hat{\theta}^* = \arg \min_{\hat{\theta}} = \mathbb{E}_t \left\{ \hat{\lambda}(t) \mathbb{E}_{\hat{\boldsymbol{X}}_0} \mathbb{E}_{\hat{\boldsymbol{X}}_t | \hat{\boldsymbol{X}}_0} \left[ \left\| \boldsymbol{r}_{\hat{\theta}}(\hat{\boldsymbol{X}}_t, t) - \nabla_{\hat{\boldsymbol{X}}_t} \log q_{0t}(\hat{\boldsymbol{X}}_t | \hat{\boldsymbol{X}}_0) \right\|^2 \right] \right\}. \tag{10}$$

So far we show a duality between the wavelet and spatial domains in score-based generative modeling. We provide a more detailed derivation in Appendix A.3. In the following sections, we will discuss the characteristics of high-frequency and low-frequency coefficients within the wavelet domain separately. For clarity in exposition, we will use $x_L$ and $x_H$ to represent the low-frequency and high-frequency coefficients in the wavelet domain, respectively, instead of using $\hat{X}$.

### 3.2.2 WELL-CONDITIONING PROPERTY OF LOW-FREQUENCY COEFFICIENTS

A signal $\boldsymbol{x} = x[n]$ with index $n$ can be decomposed as low-frequency and high-frequency coefficients. Low-frequency coefficients, often referred to as the "base" or "smooth" components, capture the primary structures and general trends within the data:

$$\boldsymbol{x}_L^k[l] = (H^k x)[l] = \sum_{n=0}^{N-1} f[n] \cdot \varphi_{k,l}[n], \tag{11}$$

where $\varphi_{k,l}[n]$ is the discrete scaling function. The scale index $k$ and the shift parameter $l$ define the analysis and reconstruction of signals at different scales and positions. $H$ is the low-pass operator.

Within the low-frequency subbands, wavelet decomposition applies the whitening effect to coefficients and concentrates the majority of energy. Therefore, the distribution of low-frequency coefficients results in a more stable and well-conditioned distribution close to a Gaussian distribution.

We provide a detailed explanation in A.4, with accompanying experiments illustrated in Fig. 5. Due to the convolution with a smooth, averaging function $\varphi$ and the concentration of energy, the covariance matrix behaves closer to a diagonal matrix and has relatively uniform eigenvalues after scaling, which indicates a lower condition number.

In Fig. 5, we illustrate the Kullback-Leibler (KL) divergence between the sample distribution and the standard Gaussian distribution (refer to A.4, A.10 for more details), demonstrating approximate log-linear decreasing with respect to the wavelet decomposition scale. This trend effectively confirms the whitening effect of wavelet decomposition to low-frequency coefficients. The whitened low-frequency wavelet coefficient distribution and lower condition number make it more suitable for SGM based on Gaussian-diffusion assumptions.

### 3.2.3 Sparsity of High-frequency coefficients

The high-frequency coefficients often represent rapid changes or transitions, which are sparse in natural images and signals. The sparsity in the high-frequency band primarily stems from the inherent characteristics of the data: most of the vital information and content (such as main features or structures) are concentrated in the low-frequency band, while the high-frequency band captures fine textures and contrastive edges. For a given shift parameter $l$, the high-frequency signal can be represented in discrete form as:

$$\boldsymbol{x}_H^k[l] = (G^k x)[l] = \sum_{n=0}^{N-1} x[n] \cdot \psi_{k,l}[n],$$ (12)

where $\psi_{k,l}[n]$ represents the discrete wavelet function and $G$ is the high-pass operator.

It is well known that natural images intrinsically present distinct distribution patterns than Gaussian distributions: in fact, the wavelet coefficient distribution of natural images is practically modeled as Generalized Laplacian, Gaussian scale mixture or Generalized Gaussian scale mixture Simoncelli (1999); Wainwright & Simoncelli (1999); Gupta et al. (2018); Buccigrossi & Simoncelli (1999). While the whitening effect is applied to the low-frequency coefficients, the high-frequency subbands in each scale retain the original distribution's long-tail and peak features. Consequently, high-frequency coefficients are usually sparse and highly correlated with their neighbors Gupta et al. (2018). Please refer to A.5 for more details.

As shown in Fig. 6, we summarize the sparsity of high-frequency subbands of CelebA-HQ images at different scales. Noticeably, high-frequency subbands at all scales exhibit strong sparsity, and the subbands at deeper scales have fewer non-zero components. Besides, some of their non-zero components are low-magnitude noise that can be easily filtered out with a small threshold, corresponding to the efficiency of traditional wavelet denoising. These observations effectively validate the highly non-Gaussian distribution of high-frequency wavelet coefficients.

## 4 Wavelet Multi-scale generative model

In this section, we first introduce the multi-scale factorization of the generative model in wavelet domain. Then, we separately build a SGM model at the coarsest scale and the multi-scale GANO for successive wavelet scales. The integration of the two generative models effectively increases sampling efficiency while maintaining good sample diversity.

### 4.1 Multi-scale Factorization in Wavelet Domain

The multi-scale image generation in the wavelet domain factorizes the probability of target image $p(\boldsymbol{x}^0)$ as below:

$$p(\boldsymbol{x}^0) = \Pi_{k=1}^S p(\boldsymbol{x}_H^k | \boldsymbol{x}_L^k) p(\boldsymbol{x}_L^S).$$ (13)

Notice the LL subband at a finer scale $k$ is jointly determined by the low-frequency and high-frequency subbands at scale $k + 1$, i.e.,

$$\boldsymbol{x}_L^k = A^T (\boldsymbol{x}_H^{k+1}, \boldsymbol{x}_L^{k+1})^T.$$ (14)

Here, $p(\boldsymbol{x}_L^S)$ can be effectively approximated through the reverse diffusion process due to the whitening effect of multi-scale wavelet decomposition. On the other hand, the conditional probability of

high-frequency subbands on the LL subband $p(\boldsymbol{x}_H^k|\boldsymbol{x}_L^k)$ is often a long-tail and peaked multimodal distribution and differs from simple Gaussian models considerably (refer to A.5). It is worth noting that a more precise and detailed depiction of this conditional distribution can be derived using the Gaussian scale mixture modeling the joint distribution of wavelet coefficients and the Gaussian-like distribution of the LL subband, though in this work, we don't rely on the complicated explicit expression of the conditional distribution but approximate it implicitly through adversarial learning.

## 4.2 SCORE-BASED GENERATIVE MODEL IN THE COARSEST WAVELET LAYER

As introduced in Ho et al. (2020); Song et al. (2020); Song & Ermon (2019), a deep neural network $s_\theta$ approximates the score function in the reverse process. Following Eq. 10, the loss function for the training of $s_\theta$ is

$$\mathbb{E}_t \left\{ \lambda(t) \mathbb{E}_{\boldsymbol{x}_L^S} \mathbb{E}_{\boldsymbol{x}_{L,t}^S|\boldsymbol{x}_L^S} \left[ \left\| \boldsymbol{s}_\theta(\boldsymbol{x}_{L,t}^S, t) - \nabla_{\boldsymbol{x}_{L,t}^S} \log q_{0t}(\boldsymbol{x}_{L,t}^S|\boldsymbol{x}_L^S) \right\|^2 \right] \right\}. \tag{15}$$

This SGM learns the reverse diffusion process at the coarsest level, which resolves the ill-positionedness of the general diffusion process in the spatial domain. Therefore, the training can be more efficient and faster than directly training on full-resolution images. After training, similar to Eq. 3, the reverse process is described as:

$$\boldsymbol{x}_{L,t-1}^S = \boldsymbol{x}_{L,t}^S + \Delta t \left( \boldsymbol{x}_{L,t}^S + 2 s_\theta(\boldsymbol{x}_{L,t}^S, t) \right) + \sqrt{2\Delta t} \mathbf{z}_t. \tag{16}$$

## 4.3 MULTI-SCALE GENERATIVE ADVERSARIAL NEURAL OPERATOR

Considering the marginal distribution of high-frequency wavelet subbands and its spatial correlation with the LL subband, we employ Generative Adversarial Network (GAN)Goodfellow et al. (2020) to model the conditional distribution of $\bar{x}_k$ on $x_k$. GAN has long been used to generate photo-realistic images rapidly, and recent work demonstrates the integration of GAN and diffusion models for improved sampling speed and mode coverage due to the capability of GAN to learn complicated multimodal distributions Creswell et al. (2018); Gui et al. (2021); Özbey et al. (2023); Wang et al. (2022); Xiao et al. (2021); Zheng et al. (2022).

As illustrated in Fig. 7, we introduce a multi-scale generative adversarial neural operator (GANO) that learns the transformation from LL subband to high-frequency subbands at various wavelet scales. The generator $G$ consists of a 5-level U-Net enhanced with attention gates Ronneberger et al. (2015); Oktay et al. (2018). The training loss for multi-resolution super-resolution GAN is

$$\mathscr{L}_G = \sum_{k=1}^{S} \left[ \lambda(G(\boldsymbol{x}_L^k, \boldsymbol{z}^k) - \boldsymbol{x}_H^k)^2 + \nu(1 - \text{SSIM}(G(\boldsymbol{x}_L^k, \boldsymbol{z}^k), \boldsymbol{x}_H^k)) - \alpha D(G(\boldsymbol{x}_L^k, \boldsymbol{z}^k)) \right], \tag{17}$$

$$\mathscr{L}_D = \sum_{k=1}^{S} \left( D(G(\boldsymbol{x}_L^k, \boldsymbol{z}^k)) - D(\boldsymbol{x}_H^k) \right). \tag{18}$$

Here $\boldsymbol{z}^k$ refers to random white noise at scale $k$, $\text{SSIM}(\cdot, \cdot)$ is the structural similarity index measure Wang et al. (2004), and $D$ is the discriminator. We penalize the generator and discriminator on the Wasserstein distances between fake and real images Arjovsky et al. (2017). The discriminator has 5 convolutional residual blocks, each halving the spatial dimension of features, and 2 fully connected layers that produce the scalar score of the input image. The architectures of the generator and discriminator are detailed in Fig. 7. Through the multi-scale operator learning in the wavelet domain and sharing the parameters across various scales; the multi-scale GANO allows a significant reduction in the number of trainable parameters while maintaining comparable performance to existing methods Guth et al. (2022b).

**Multi-scale operator Learning.** Recent work has witnessed the innovative integration of wavelet analysis and deep learning to achieve advanced operator learning Tripura & Chakraborty (2022); Gupta et al. (2021); Xiao et al. (2022). This approach primarily involves mapping coefficients within the wavelet domain, facilitating function-to-function mapping. The operator learning work as presented in Li et al. (2020); Gupta et al. (2021) demonstrates super-resolution properties through constructing mappings in the frequency domain.

**One-shot GANO** To avoid unstable training and mode collapse of GAN Kodali et al. (2017); Gui et al. (2021), related works break the SGM sampling process into multiple denoising steps and accelerate the time-consuming reverse process with GAN Wang et al. (2022); Xiao et al. (2021). In contrast to these studies where GAN learns the complicated sample distribution $p(\boldsymbol{x}^0)$ from standard Gaussian directly, the multi-scale GANO in our framework generates samples in one shot instead of iterative, progressive sampling. This is benefited from learning a simpler conditional distribution $p(\boldsymbol{x}_H^k|\boldsymbol{x}_L^k)$ where the low-frequency condition of $\boldsymbol{x}_L^k$ provides an excellent reference to the target, simplifies the task and stabilizes the training. In addition, the SGM in LL subband in our method improves mode coverage and diversity of generated images than traditional GANs.

## 5 EXPERIMENTS

### 5.1 IMPLEMENTATION AND TRAINING DETAILS

For training, the input images are based on 128x128 resolution. We use Adam optimizer Kingma & Ba (2014) with the learning rate $10^{-4}$ for the diffusion model, and AdamW optimizers Loshchilov & Hutter (2017) for the generator and discriminator using learning rates of $10^{-4}$ and $10^{-5}$, respectively. The diffusion model is trained with the batch size of 64 for 50000 iterations, while the generator and discriminator are trained with the batch size of 128 for 150 epochs.

For evaluation metrics, we use the Frechet inception distance (FID) Heusel et al. (2017) to measure the image quality. The sampling time is averaged over 10 trials when generating a batch of 64 images. Training code and model weights are available for the sake of reproducibility. All tasks are conducted on a NVIDIA V100 GPU.

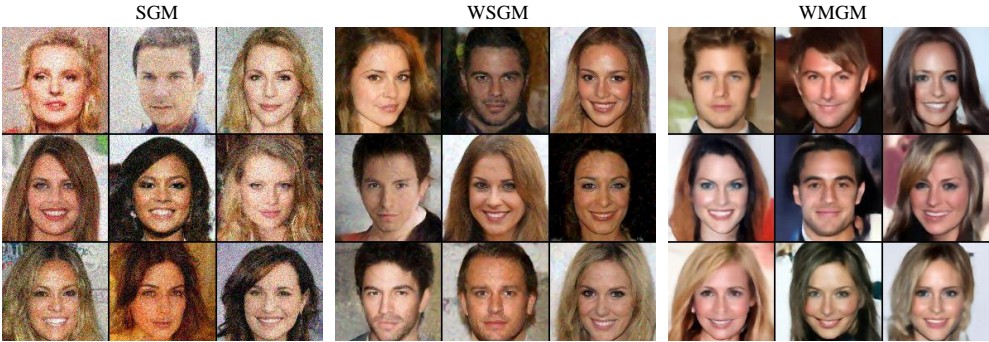

Figure 3: Generation of SGM, WSGM and our method on CelebA-HQ datasets. 16 discretization steps per scale was utilized for WSGM, and our model and SGM keep the same total sampling steps.

### 5.2 RESULTS AND COMPARISON

In this section, we provide detailed information about our experiments conducted on the CelebA-HQ dataset by Karras et al. (2017), as well as the AFHQ dataset introduced by Choi et al. (2020). The outcomes of the proposed WMGM are presented in Table 1. To ensure a fair comparison with previous methodologies Guth et al. (2022b), we separately train our model and their approach on these datasets and evaluate the performance on the same test sets. We maintain consistent settings for both CelebA-HQ and the AFHQ dataset, and employ the FID as the evaluation metric.

In Table 1, we use discretization steps of 16. AFHQ-Cat refers to the 'cat' category in AFHQ. Besides assessing generation quality with FID, we evaluate efficiency using model parameters and inference time. Our performance surpasses SGM Song et al. (2020) and WSGM Guth et al. (2022b), achieving FID scores of 30.58 on 30k CelebA data, 25.38 on 5k CelebA and 16.29 on AFHQ-Cat.

Figure 4 illustrates the FID comparison on the AFHQ-Cat and CelebA-HQ test set utilizing various total sampling steps ranging from 6 to 192. In all of these settings, we attain state-of-the-art generation quality in comparison to both SGM and WSGM. On the AFHQ-Cat dataset, our method rapidly reaches a convergence with FID around 16 in a total of 48 discretization steps, while WSGM needs significantly more discretization steps of 192 for the same FID level. This result further emphasizes the fast sampling advantage of our method over related studies.

| Methods | FID↓ | | | Parameters↓ | Sampling Time(ms)↓ |
|---|---|---|---|---|---|
| | CelebA-HQ (5K) | AFHQ-Cat (5K) | CelebA-HQ (30K) | | |
| SGM | 90.83 | 80.93 | 78.50 | 160M | 15093 |
| WSGM | 49.97 | 17.12 | 26.74 | 351M | 11097 |
| **WMGM(Ours)** | **30.58** | **16.29** | **25.38** | **89M** | **4175** |

Table 1: Comparison of SGM, WSGM and our method. 16 discretization steps per scale was utilized for WSGM, our model and SGM keep the same total sampling steps. Sampling time including all steps of each method. **Bold** data represents the optimal results overall.

With such notable performance gains, it is noteworthy that our approach exhibits exceptional efficiency in terms of model size, inference time, and data requirements. As evident in Table 1, our model encompasses merely 89M parameters, making it 44.38% smaller than SGM and 74.64% smaller than WSGM. Moreover, we observe accelerated sampling during inference as compared to those methodologies. The sampling time of the proposed WMGM is 72.33% lower than SGM and 62.38% lower than WSGM. Additionally, beyond these direct comparisons in size and time, we ascertain that our approach achieves high-quality image generation with mere 16 total sampling steps for an image of $128 \times 128$. Notably, our model converges even with a modest training dataset comprising only 5000 images, whereas both SGM and WSGM fail under such circumstances.

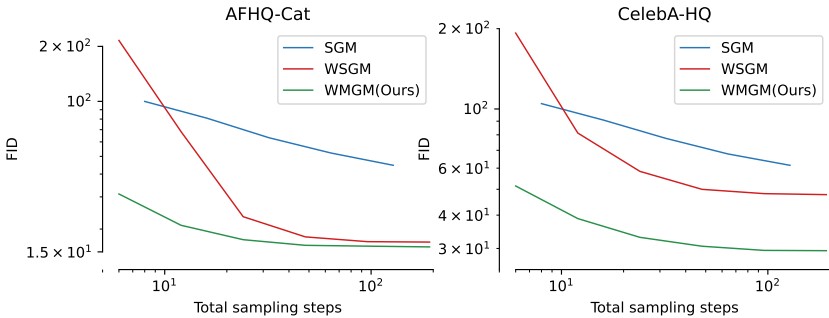

Figure 4: FID of SGM, WSGM and our method on AFHQ-Cat and CelebA-HQ datasets w.r.t. various total sampling steps.

## 6 CONCLUSION

In conclusion, our work addresses the score-based generative models' ill-conditioness in the spatial domain and challenges associated with the wavelet domain and presents the Wavelet Multi-Scale Generative Model (WMGM). We theoretically demonstrate the duality of diffusion process between the spatial and wavelet domains and comprehensively investigate the distribution of wavelet coefficients. We design an effective score-based generative model at the coarsest scale utilizing the whitening effect of low-frequency coefficients and the multi-scale GANO to handle the non-Gaussian distribution of high-frequency wavelet coefficients. Our extensive experimental results demonstrate that WMGM not only outperforms competing methods in terms of sampling speed and generation quality but also reduces trainable parameters and sampling steps. Overall, our research represents a significant step forward in diffusion modeling techniques, opening up new avenues for generative modeling in the wavelet domain.

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

# A APPENDIX

## A.1 CODE AVAILABILITY

The code is available at https://anonymous.4open.science/r/WMGM-3C47/.

## A.2 DIFFUSION IN THE WAVELET DOMAIN

We first investigate the effect of noise evolution in the spatial domain. In our extensive image perturbation analysis, we observed the ramifications of incrementally introduced Gaussian noise in the spatial domain. Initially, the superimposed Gaussian noise manifests predominantly as high-frequency perturbations. With an increase in the strength and duration of noise injection, these perturbations start to mask the primary structures of the original image, causing the entire image to be progressively characterized by Gaussian noise attributes. This renders the image increasingly homogeneous, dominated by high-frequency disturbances.

The influence of noise evolution extends beyond the spatial domain, and its progressive perturbation to different frequency bands of the image can be better understood from a wavelet domain perspective. When the perturbed images are subjected to a wavelet transform, we noted a systematic series of effects:

1. **Concentration of High-Frequency Effects:** Recall the power spectra of natural images following the $\frac{1}{f^2}$ decaying rule Field (1987); Van der Schaaf & van Hateren (1996) and the constant power spectra of Guassian noise. Therefore, as noise is introduced, it predominantly manifests in the high-frequency subbands (LH, HL, HH). This is due to the wavelet transform's ability to separate out high-frequency details from the low-frequency approximations of an image.

2. **Cumulative Effect:** As more noise is added, it not only augments the existing high-frequency perturbations but also starts to influence the low-frequency content, especially as the strength and/or duration of the noise becomes significant. Gradually, the noise begins to leave its footprint in the low-frequency subband (LL) as the original low-frequency content gets progressively masked or drowned by the noise.

3. **Multi-Scale Structure of Wavelets:** The wavelet transform possesses a multi-resolution analysis characteristic. In the initial wavelet decomposition, the LL subband still contains most of the energy and primary information of the image, while the LH, HL, and HH subbands capture finer details. As the amount of introduced noise reaches a certain threshold, these fine details and the main structure of the original image get masked by the noise, leading to a more uniform distribution of energy across both low and high-frequency subbands. Over time, the coefficient distribution in the LL subband begins to resemble a Gaussian distribution more closely.

In summary, the gradual introduction of noise first impacts the high-frequency subbands and, as noise accumulates, the low-frequency subbands are also affected. When the noise level is sufficiently high, the entire image becomes dominated by the Gaussian noise, whether in the spatial or wavelet domain.

## A.3 DUALITY OF THE DIFFUSION PROCESS IN THE SPTIAL SPACE AND THE WAVELET DOMAIN

### A.3.1 FORWARD PROCESS

To simplify the notations, let $x$ be an image vector, and we can write the discrete wavelet transform (DWT) as:
$$\hat{X} = AX, \quad X \in \mathbb{R}^d.$$

Here, $A$ is the discrete wavelet matrix. This matrix is an orthogonal matrix, i.e., $AA^\top = I$. Several choices of $A$ are widely applied, such as Haar wavelets.

For the score-based generative modeling process, we consider the forward/noising process. This process can be mathematically formulated as the Ornstein–Uhlenbeck (OU) process. The general

time–rescaled OU process can be written as

$$dX_t = -g(t)^2 X_t dt + \sqrt{2} g(t) dB_t. \tag{19}$$

Here, $B_t$ is a standard d–dimensional Brownian motion. We perform DWT to $X_t$ and figure out $\hat{X}_t$ also observes the same OU process.

$$d\hat{X}_t = -g(t)^2 \hat{X}_t dt + \sqrt{2} g(t) A dB_t, \quad \hat{X}_0 = A X_0. \tag{20}$$

Let $\hat{B}_t = A B_t$, $\hat{B}_t$ is also a standard Brownian motion. We let $X_0$ be sampled from distribution $p$. Then $\hat{X}_0$ is from the distribution

$$q = \mathcal{T}_A \# p. \tag{21}$$

Here, $\mathcal{T}_A$ is the $A$ linear transform operation. $\#$ is the pushforward operation, which gives

$$q(x) = p(A^\top x). \tag{22}$$

Let $p_t$ be the density distribution of $X_t$, $q_t$ be the density distribution of $\hat{X}_t$. We have

$$q_t = \mathcal{T}_A \# p_t, \quad q_t(x) = p_t(A^\top x). \tag{23}$$

Let

$$s_t = \nabla \log p_t, \quad r_t = \nabla \log q_t. \tag{24}$$

be the score functions of two processes. We have

$$r_t(x) = \frac{\nabla q_t(x)}{q_t(x)} = \frac{A \nabla p_t(A^\top x)}{p_t(A^\top x)} = A s_t(A^\top x). \tag{25}$$

### A.3.2 DENOISING/REVERSE PROCESS

We use $X_t^\leftarrow$ and $\hat{X}_t^\leftarrow$ to denote the reverse process. With the common assumption that $g(t) = 1$ in standard diffusion models, the reverse processes follow:

$$\begin{aligned}
dX_t^\leftarrow &= \left(X_t^\leftarrow + 2 s_{T-t}(X_t^\leftarrow)\right) dt + \sqrt{2} dB_t, \\
d\hat{X}_t^\leftarrow &= \left(\hat{X}_t^\leftarrow + 2 r_{T-t}(\hat{X}_t^\leftarrow)\right) dt + \sqrt{2} d\hat{B}_t.
\end{aligned} \tag{26}$$

Here, $\hat{B}_t = A B_t$. We look into the second SDE.

$$\begin{aligned}
d\hat{X}_t^\leftarrow &= \left(\hat{X}_t^\leftarrow + 2 r_{T-t}(\hat{X}_t^\leftarrow)\right) dt + \sqrt{2} d\hat{B}_t \\
&= \left(\hat{X}_t^\leftarrow + 2 A s_{T-t}(A^\top \hat{X}_t^\leftarrow)\right) dt + \sqrt{2} d\hat{B}_t \\
A^\top d\hat{X}_t^\leftarrow &= \left(A^\top \hat{X}_t^\leftarrow + 2 s_{T-t}(A^\top \hat{X}_t^\leftarrow)\right) dt + \sqrt{2} A^\top d\hat{B}_t.
\end{aligned} \tag{27}$$

Replacing $A^\top \hat{X}_t^\leftarrow$ by $X_t^\leftarrow$. We can get back to the first equation. The training processes for $s_\theta, r_{\hat{\theta}}$ with $X_t^{(i)}, \hat{X}_t^{(i)}$ also following the same standard denoising score matching loss function as follows:

$$\mathbb{E}_t \left\{ \lambda(t) \mathbb{E}_{X_0} \mathbb{E}_{X_t|X_0} \left[ \|s_\theta(X_t, t) - \nabla_{X_t} \log p_{0t}(X_t|X_0)\|^2 \right] \right\}$$

$$\mathbb{E}_t \left\{ \hat{\lambda}(t) \mathbb{E}_{\hat{X}_0} \mathbb{E}_{\hat{X}_t|\hat{X}_0} \left[ \left\| r_{\hat{\theta}}(\hat{X}_t, t) - \nabla_{\hat{X}_t} \log q_{0t}(\hat{X}_t|\hat{X}_0) \right\|^2 \right] \right\}. \tag{28}$$

The forward and reverse probability distribution function $p_{0t}$ and $q_{0t}$ are defined following the standard SGM model, in other words,

$$\begin{aligned}
p_{0t}(X_t|X_0) &= \mathcal{N}(X_t; \sqrt{\bar{\alpha}_t} X_0, (1 - \bar{\alpha}_t) I), \\
q_{0t}(\hat{X}_t|\hat{X}_0) &= \mathcal{N}(\hat{X}_t; \sqrt{\bar{\alpha}_t} \hat{X}_0, (1 - \bar{\alpha}_t) I).
\end{aligned} \tag{29}$$

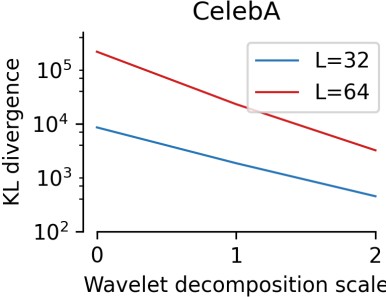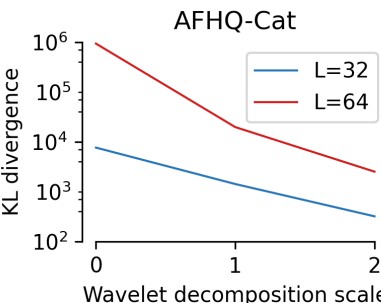

Figure 5: KL divergence of sample distribution (scale = 0) and LL coefficient distributions (scale = 1,2) to standard Gaussian distribution. Images were downsampled to size $L \times L$ before wavelet decomposition.

### A.4 GAUSSIAN TENDENCY OF LOW-FREQUENCY COEFFICIENTS IN HIGHER SCALES

In this section, we first experimentally showcase the KL divergence between sample distribution of low-frequency coefficients and standard Gaussian distribution on CelebA-HQ and AFHQ-Cat datasets. 2-scale wavelet decomposition was implemented to each image, and the sample mean and covariance were calculated accordingly to the raw image (scale 0) and LL subbands at scale 1 and 2. The KL divergence of sample distribution to standard Gaussian is detailed in A.10.

Next, we provide theoretical explanation to this whitening effect. Consider $X$ as a random variable representing the intensity of an image pixel, with a distribution characterized by mean $\mu$ and standard deviation $\sigma$. Given $n$ observations of $X$, these are not strictly independent but exhibit a dependence due to the spatial proximity in the image.

The dependence of the pixels varies in different situations. For a smooth clarification, let's denote the dependence between two pixels separated by a distance $d$ as $C(d)$. Apparently, the dependence will decrease with the increase of $d$. We smoothly assume the dependence of two pixels can be represented as a power-law decay:

$$C(d) = \frac{1}{(1 + \alpha d)^\beta} \tag{30}$$

Here, $\alpha$ and $\beta$ are parameters controlling the rate of decay in correlation or dependence with distance, and these parameters would need to be estimated empirically from the image data. The distance $d$ between pixels can be calculated using a suitable distance metric, such as Euclidean distance, depending on the data's specific characteristics.

When the image undergoes the decomposition processes such as low-pass filtering and downsampling, this dependence $C(d)$ diminishes, particularly for larger $d$. Mathematically, this diminishing dependence can be represented with a function that decreases as $d$ increases. Given the ubiquitous noise in natural images, resonably we can assume the dependence $C(d)$ is truncated beyond $d_{max}$.

Consider $X$ as a random variable representing the intensity of an image pixel, with a general distribution, mean $\mu$, and standard deviation $\sigma$. Given $n$ observations of $X$, these are not necessarily independent but possess a dependence structure (for instance, due to the spatial proximity of pixels in an image), we define:

$$\bar{X} = \frac{1}{n}(X_1 + X_2 + \ldots + X_n), \tag{31}$$

When the image undergoes low-pass filtering and down-sampling, the dependence between adjacent pixels is further weakened according to Eq. 31. In the multiscale analysis context, the low-frequency components at scale $k$ can be approximated as the average intensity within the corresponding local region. As scale $k$ increases, each local region incorporates more pixels, and the average is taken over $n_k$ pixel values. According to the General Central Limit Theorem Billingsley (2013); Rosenblatt (1956), given the weak dependence structure and the law of large numbers for weakly dependent sequences, as $n_k$ approaches infinity, the distribution of $\bar{X}_k$ will converge to a normal distribution with mean $\mu$ and standard deviation $\frac{\sigma}{\sqrt{n_k}}$:

$$\bar{X}_k \xrightarrow{\mathrm{d}} \mathcal{N}(\mu, \frac{\sigma^2}{n_k}), \quad \text{as } n_k \to \infty \tag{32}$$

Besides, as $\sqrt{n_k} \gg d_{max}$, the correlation between adjacent pixels at scale $k$ will diminish and can be regarded independent, identically distributed variables.

Therefore, at larger scales where $n_k$ is sufficiently large, the low-frequency coefficients' distribution increasingly resembles a Gaussian distribution. This explains why, in multiscale analysis, signal components at larger scales (i.e., lower frequencies) tend to be normally distributed.

However, in practical applications, achieving infinity for $n_k$ is infeasible due to constraints imposed by the real-world data and computational resources. Moreover, the border effect, which is a common issue encountered during the wavelet transform process Jensen & la Cour-Harbo (2001), imposes additional limitations. This necessitates a thoughtful tradeoff in selecting an appropriate coarsest layer for analysis. In the context of our paper, we choose a layer size of 32 to effectively navigate through this tradeoff. Notwithstanding these practical constraints, the inherent Gaussian tendency of low-frequency coefficients at higher scales still plays a pivotal role. It ensures a better-conditioned Stein score during the diffusion process in the low-frequency wavelet domain.

### A.5 SPARSE TENDENCY OF OF HIGH-FREQUENCY WAVELET COEFFICIENTS

By examining the statistical sparsity of images in the CelebA-HQ dataset, we show that the distribution of high-frequency wavelet coefficients is highly non-Gaussian. For a given image $\boldsymbol{x}$ and threshold $t$, the sparsity of its high-frequency coefficients at $k$-scale is defined as:

$$s(\boldsymbol{x}_H^k) = \frac{\|\mathbf{1}\{\boldsymbol{x}_H^k \leq t\}\|}{L^2}, \ k = 1, 2, \cdots \tag{33}$$

Here $\|\cdot\|$ is the norm counting the number of 1s in the vector. In this way, we could estimate the expected sparsity of the true marginal distribution $p(\boldsymbol{x}_H^k)$. Considering that the LL coefficients with approximate Gaussian distribution given the whitening effect of wavelet decomposition, we have the following proposition.

**Proposition 1.** *For a sufficiently large $k$, if the expected sparsity of $\boldsymbol{x}_H^k$ has a lower bound $\alpha$*

$$\mathbb{E}(s(\boldsymbol{x}_H^k)) \geq \alpha, \tag{34}$$

*where $\alpha \in [0, 1]$. Then the conditional expected sparsity of $\boldsymbol{x}_H^k$ on $\boldsymbol{x}_L^k$ is bounded by*

$$\mathbb{E}(s(\boldsymbol{x}_H^k)|\boldsymbol{x}_L^k) \geq \alpha - \varepsilon, \tag{35}$$

*where $\varepsilon > 0$ is a small positive number determined by $k$.*

*Proof.* According to A.4, for a sufficiently large $k$ we could assume that

$$\int |p(\boldsymbol{x}_L^k) - f_k(\boldsymbol{x}_L^k)| d\boldsymbol{x}_L^k \leq \varepsilon, \tag{36}$$

where $f_k(\boldsymbol{x}_L^k)$ is the PDF of standard Gaussian distribution. Notice that

$$\mathbb{E}(s(\boldsymbol{x}_H^k)) = \iint s(\boldsymbol{x}_H^k) p(s(\boldsymbol{x}_H^k)|\boldsymbol{x}_L^k) p(\boldsymbol{x}_L^k) d\boldsymbol{x}_L^k ds \tag{37}$$

$$= \int \mathbb{E}(s(\boldsymbol{x}_H^k)|\boldsymbol{x}_L^k) p(\boldsymbol{x}_L^k) d\boldsymbol{x}_L^k \geq \alpha \tag{38}$$

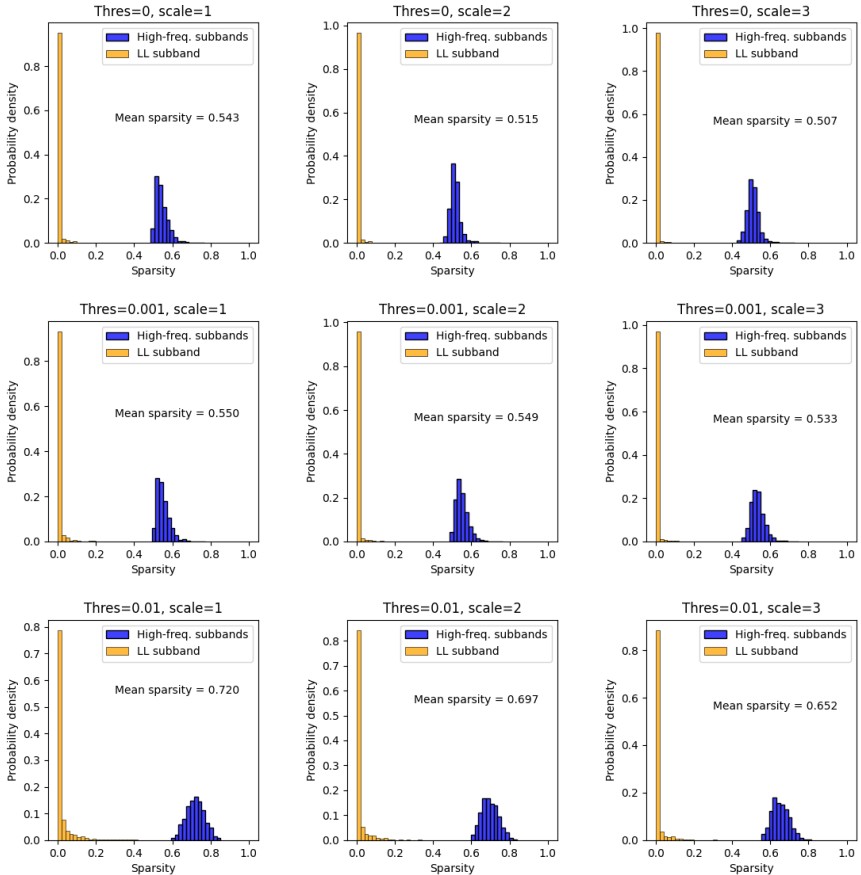

Figure 6: High frequency coefficient sparsity of CelebA-HQ images. Low-magnitude coefficients smaller than Thres is recognized as white noise and filtered to zero.

Since $s$ is a bounded function in $[0, 1]$, $\mathbb{E}(s(\boldsymbol{x}_H^k)|\boldsymbol{x}_L^k)$ has an uniform lower bound with respect to all $\boldsymbol{x}_L^k$, denoted as $\alpha'$. In other words, there exists $\alpha' \in [0, 1]$ such that

$$\mathbb{E}(s(\boldsymbol{x}_H^k)|\boldsymbol{x}_L^k) \geq \alpha', \; \forall \boldsymbol{x}_L^k \tag{39}$$

We can get

$$\int \mathbb{E}(s(\boldsymbol{x}_H^k)|\boldsymbol{x}_L^k)p(\boldsymbol{x}_L^k)d\boldsymbol{x}_L^k \tag{40}$$

$$= \int \mathbb{E}(s(\boldsymbol{x}_H^k)|\boldsymbol{x}_L^k)f_k(\boldsymbol{x}_L^k)d\boldsymbol{x}_L^k + \int \mathbb{E}(s(\boldsymbol{x}_H^k)|\boldsymbol{x}_L^k)(p(\boldsymbol{x}_L^k) - f_k(\boldsymbol{x}_L^k))d\boldsymbol{x}_L^k \tag{41}$$

$$\geq \alpha' \tag{42}$$

Similarly, it is easy to see that $1$ is a trivial uniform upper bound for $\mathbb{E}(s(\boldsymbol{x}_H^k)|\boldsymbol{x}_L^k)$. Thus,

$$\mathbb{E}(s(\boldsymbol{x}_H^k)|\boldsymbol{x}_L^k) \geq \alpha' = \int \mathbb{E}(s(\boldsymbol{x}_H^k)|\boldsymbol{x}_L^k)p(\boldsymbol{x}_L^k)d\boldsymbol{x}_L^k - \varepsilon \geq \alpha - \varepsilon. \tag{43}$$

$\square$

Consequently, we can see the conditional distribution of $\boldsymbol{x}_H^k$ on $\boldsymbol{x}_L^k$ exhibits highly non-Gaussian properties and yields sparse samples.

## A.6 ATTENTION-GAN ARCHTECTURE DESCRIPTION

As shown in Figure 7, the core of the generator is based on the UNet structure, and it also incorporates attention modules during the upsampling process. After data enters the generator, it first goes through five layers of downsampling followed by five layers of upsampling. With each upsampling, the data subsequently passes through an attention module. After completing these operations, the data finally passes through two convolutional operations and activation functions to output the required values.

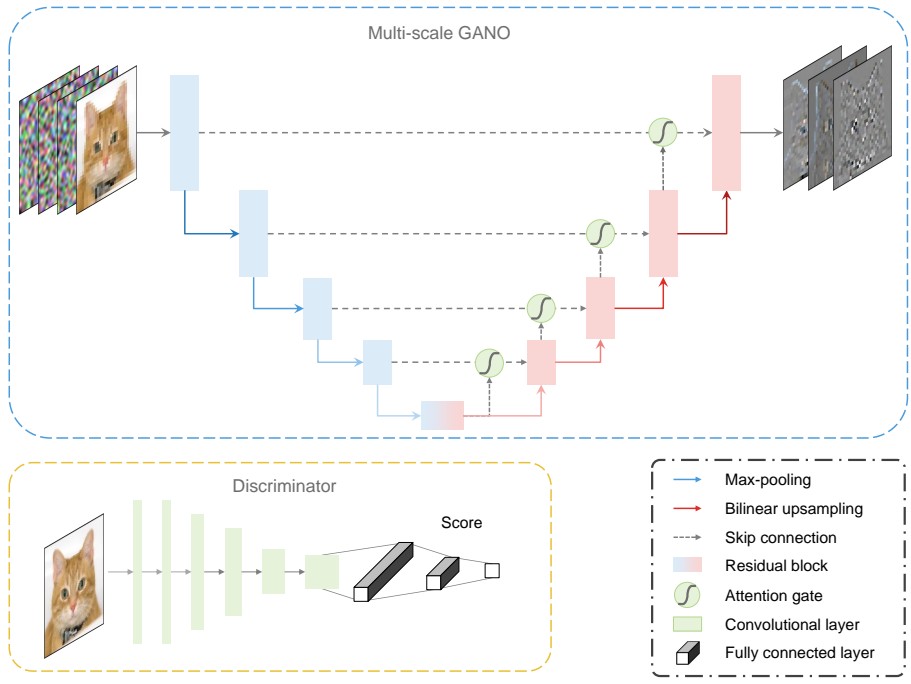

Figure 7: Network architecture of the proposed multi-scale GANO.

The total number of parameters in the Attention-GAN model is approximately 36 million, with a model depth of 106 layers. The model's width starts from the size of the input channel, goes through a downsampling process to a maximum of 1024, and then becomes the size of the output channel after the upsampling and attention process. The width of the intermediate layers can be controlled by adjusting the parameters. The skip connections in this model are primarily characterized by the use of corresponding downsampling information during the upsampling process. The difference lies in that the data, after being processed by the upsampling layers, serve as input information for the attention gates and are entered into the attention block together with the corresponding information from the downsampling process. The implementation of attention gate is as follows.

- It first applies the transformations 'self.W_g' and 'self.W_x' to the gating signal 'g' and the local signal 'x' respectively, producing two intermediate feature maps.

- These feature maps are then added together and passed through a ReLU activation function and a 2D Convolutional layer to generate the attention map.

- The attention map is element-wise multiplied by the original local signal 'x' to produce the output of the attention block, which selectively emphasizes certain features of 'x' based on the gating signal 'g'.

For more information, please refer to the open-sourced codes of our method.

A.7  WAVELET TRANSFORM

The wavelets represent sets of functions that result from dilation and translation from a single function, often termed as 'mother function', or 'mother wavelet'. For a given mother wavelet $\psi(x)$, the resulting wavelets are written as

$$\psi_{a,b}(x) = \frac{1}{|a|^{1/2}} \psi\left(\frac{x-b}{a}\right), \qquad a, b \in \mathbb{R}, a \neq 0, x \in D, \tag{44}$$

where $a, b$ are the dilation, translation factor, respectively, and $D$ is the domain of the wavelets under consideration. In this work, we are interested in the compactly supported wavelets, or $D$ is a finite interval $[m, n]$, and we also take $\psi \in L^2$. For the rest of this work, without loss of generality, we restrict ourself to the finite domain $D = [0, 1]$, and extension to any $[m, n]$ can be simply done by making suitable shift and scale.

A.7.1  1D DISCRETE WAVELET TRANSFORM (1D DWT)

**Multiresolution Analysis (MRA)**  The Wavelet Transform employs Multiresolution Analysis (MRA) as a fundamental mechanism. The basic idea of MRA is to establish a preliminary basis in a subspace $V_0$ of $L^2(\mathbb{R})$, and then use simple scaling and translation transformations to expand the basis of the subspace $V_0$ into $L^2(\mathbb{R})$ for analysis on multiscales.

For $k \in \mathbb{Z}$ and $k \in \mathbb{N}$, the space of scale functions is defined as: $\mathbf{V}_k = \{f | \text{the restriction of } f \text{ to the interval } (2^{-k}l, 2^{-k}(l+1)), \text{ for all } l = 0, 1, \ldots, 2^k - 1, \text{ and f vanishes elsewhere}\}$. Therefore, the space $\mathbf{V}_k$ has dimension $2^k$, and each subspace $\mathbf{V}_i$ is contained in $\mathbf{V}_{i+1}$ shown as

$$\mathbf{V}_0 \subset \mathbf{V}_1 \subset \mathbf{V}_2 \subset \ldots \mathbf{V}_n \subset \ldots. \tag{45}$$

Given a basis $\varphi(x)$ of $\mathbf{V}_0$, the space $\mathbf{V}_k$ is spanned by $2^n$ functions obtained from $\varphi(x)$ by shifts and scales as

$$\varphi_l^k(x) = 2^{k/2} \varphi(2^k x - l), \quad l = 0, 1, \ldots, 2^k - 1. \tag{46}$$

The functions $\varphi_l(x)$ are also called scaling functions which can project a function to the approximation space $\mathbf{V}_0$. The wavelet subspace $\mathbf{W}_k$ is defined as the orthogonal complement of $\mathbf{V}_k$ in $\mathbf{V}_{k+1}$, such that:

$$\mathbf{V}_k \bigoplus \mathbf{W}_k = \mathbf{V}_{k+1}, \quad \mathbf{V}_k \perp \mathbf{W}_k; \tag{47}$$

and $\mathbf{W}_k$ has dimension $2^k$. Therefore, the decomposition can be obtained as

$$\mathbf{V}_k = \mathbf{V}_0 \bigoplus \mathbf{W}_0 \bigoplus \mathbf{W}_1 \ldots \bigoplus \mathbf{W}_{k-1}. \tag{48}$$

To form the orthogonal basis for $\mathbf{W}_k$, the basis is constructed for $L^2(\mathbb{R})$. The bases can be obtained by translating and dilating from the wavelet function which is orthogonal to the scaling function. The wavelets function $\psi(x)$ shown as follows:

$$\psi_l^k(x) = 2^{k/2} \psi(2^k x - l), \quad l = 0, 1, \ldots, 2^k - 1. \tag{49}$$

where the wavelet function $\psi(x)$ are orthogonal to low-order functions (vanishing moments), here is an example for first-order polynomial:

$$\int_{-\infty}^{\infty} x\psi_j(x)dx = 0. \tag{50}$$

Wavelets' properties of orthogonality and vanishing moments are pivotal for effective data representation. Orthogonality ensures each wavelet coefficient distinctly captures specific data features without overlap. In contrast, the vanishing moments allow wavelets to efficiently encapsulate smooth polynomial trends in the data, facilitating both clear approximations and feature extraction. Together, these attributes make wavelets adept at concise and unambiguous data analysis.

**Wavelet Decomposition**  The Discrete Wavelet Transform (DWT) offers a multi-resolution analysis of signals, gaining popularity in signal processing, image compression, and numerous other domains.

For a discrete signal $f[n]$, we can obtain its approximation coefficients $cA$ and detail coefficients $cD$ using the scaling function $\phi(x)$ and the wavelet function $\psi(x)$, respectively. This is done by convolving $f[n]$ with the respective filters followed by downsampling:

$$cA[k] = \sum_n h[n - 2k]f[n] \qquad\qquad cD[k] = \sum_n g[n - 2k]f[n] \qquad (51)$$

Where $h[n]$ and $g[n]$ denote the low-pass and high-pass filters respectively. The decomposition process can be recursively applied to the approximation coefficients $cA$ for deeper multi-level decompositions. Each subsequent level reveals coarser approximations and finer details of the signal.

**Wavelet Reconstruction** Upon having decomposed a signal using the DWT, reconstruction aims to rebuild the original signal from its wavelet coefficients. This process uses the inverse wavelet and scaling transformations, necessitating another pair of filters known as reconstruction or synthesis filters.

For a given set of approximation coefficients $cA$ and detail coefficients $cD$, the reconstructed signal $f'[n]$ can be obtained using:

$$f'[n] = \sum_k cA[k] \cdot h_0[n - 2k] + cD[k] \cdot g_0[n - 2k] \qquad (52)$$

Here, $h_0[n]$ and $g_0[n]$ are the synthesis filters related to the decomposition filters $h[n]$ and $g[n]$. Typically, these synthesis filters are closely tied to the decomposition filters, often being their time-reversed counterparts.

For multi-level decompositions, the reconstruction process is conducted in a stepwise manner. Starting from the coarsest approximation, it's combined with the detail coefficients from the highest level. This resultant signal then acts as the approximation for the next level, and the procedure is iteratively repeated until the finest level is reached, thus completing the reconstruction.

### A.7.2 2D DISCRETE WAVELET TRANSFORM (2D DWT)

The 2D DWT is a pivotal technique for image analysis. Unlike the traditional Fourier Transform which primarily provides a frequency view of data, the 2D DWT offers a combined time-frequency perspective, making it especially apt for analyzing non-stationary content in images.

For a given image $I(x, y)$, its wavelet transform is achieved using a pair of filters: low-pass and high-pass, followed by a down-sampling operation. The outcome is four sets of coefficients: approximation coefficients, horizontal detail coefficients, vertical detail coefficients, and diagonal detail coefficients.

This transformation can be mathematically represented as:

$$\begin{aligned}
\text{Approximation Coefficients: LL} &= \text{DWT}(I(x, y) * h(x) * h(y)) \\
\text{Horizontal Detail Coefficients: LH} &= \text{DWT}(I(x, y) * h(x) * g(y)) \\
\text{Vertical Detail Coefficients: HL} &= \text{DWT}(I(x, y) * g(x) * h(y)) \\
\text{Diagonal Detail Coefficients: HH} &= \text{DWT}(I(x, y) * g(x) * g(y))
\end{aligned} \qquad (53)$$

where $h(x, y)$ and $g(x, y)$ are the low-pass and high-pass filters, respectively.

It is crucial to note that these transformations are typically followed by a down-sampling operation. The convolution with wavelet filters provides a multi-resolution representation, and the down-sampling reduces the spatial resolution, leading to the hierarchical structure characteristic of wavelet decompositions. The 2D DWT offers a comprehensive approach to understand the intricate details of images by bridging the time (spatial) and frequency domains. This representation not only simplifies image analysis but also offers unique insights unattainable through conventional frequency-only transformations.

### A.7.3 HAAR WAVELET TRANSFORM

As one of the most basic wavelets, Haar wavelet has simplicity and orthogonality, ensuring its effective implementation in digital signal processing paradigms. The straightforward and efficient

characteristics of Haar wavelets make them highly practical and popular for various applications, and their successful implementation in other works prompted us to utilize them in our project as well. When we perform wavelet decomposition on an image using the discrete wavelet transform (DWT), we can obtain an approximate representation that captures the main features or overall structure of the image, as well as finer details that capture the high-frequency information in the image. Through further multi-resolution analysis (MRA), we can view the image at different scales or levels, resulting in a view containing more detail.

The Haar wavelet and its associated scaling function can be formally delineated as:

$$\psi(t) = \begin{cases} 1 & \text{for } 0 \le t < 0.5 \\ -1 & \text{for } 0.5 \le t < 1 \\ 0 & \text{elsewhere} \end{cases}, \qquad \phi(t) = \begin{cases} 1 & \text{for } 0 \le t < 1 \\ 0 & \text{elsewhere} \end{cases} \tag{54}$$

Among them, $\psi(t)$ is the Haar wavelet function, and $\phi(t)$ is the Haar scaling function.

Haar wavelet transform has wide applications in two-dimensional space, especially in image analysis. This 2D extension retains its inherent decomposition method for 1D signals but applies it sequentially to the rows and columns of the image. With these definitions in hand, the filter coefficients are computed by evaluating the inner products. Specifically, for the Haar wavelet, we have:

- Low-pass filter coefficients (h):

$$h_0 = \int_0^1 \phi(t)\phi(2t)dt, \qquad h_1 = \int_0^1 \phi(t)\phi(2t - 1)dt. \tag{55}$$

- High-pass filter coefficients (g):

$$g_0 = \int_0^1 \psi(t)\psi(2t)dt, \qquad g_1 = -\int_0^1 \psi(t)\psi(2t - 1)dt. \tag{56}$$

For filter coefficients, usually we want them to be unitized (i.e. their L2 norm is 1). After calculation, $h_0 = \frac{1}{\sqrt{2}}$, and similarly for the other coefficients. Consequently, the Haar filter coefficients are:

$$h = \left[\frac{1}{\sqrt{2}}, \frac{1}{\sqrt{2}}\right], \qquad g = \left[\frac{1}{\sqrt{2}}, -\frac{1}{\sqrt{2}}\right]. \tag{57}$$

When extending the one-dimensional DWT to two dimensions for image processing, the filter coefficients are used in a matrix form to operate on the image. When applying the filters horizontally on rows, we consider the outer product of the filter vector with a column unit vector. Similarly, for the vertical operation down the columns, the outer product of the filter vector with a row unit vector is considered. The two-dimensional filter matrices become:

$$H = h^T \times h = \begin{bmatrix} \frac{1}{\sqrt{2}} & \frac{1}{\sqrt{2}} \\ \frac{1}{\sqrt{2}} & \frac{1}{\sqrt{2}} \end{bmatrix}, \qquad G = g^T \times g = \begin{bmatrix} \frac{1}{\sqrt{2}} & -\frac{1}{\sqrt{2}} \\ -\frac{1}{\sqrt{2}} & \frac{1}{\sqrt{2}} \end{bmatrix}. \tag{58}$$

The matrix $H$ corresponds to low-pass filtering in the horizontal and vertical directions, representing the approximate information in the image, while $G$ corresponds to the high-pass filtering in the horizontal and vertical directions, capturing the details in the image. When performing the wavelet transformation of an image using Haar wavelets, these matrices are used to derive the approximation($LL$), horizontal detail($LH$), vertical detail($HL$), and diagonal detail($HH$) coefficients.

## A.8 Score Regularity for Discretization

**Theorem 1.** *Suppose the Gaussian distribution $p = \mathcal{N}(0, \Sigma)$ and distribution $\tilde{p}_0$ from time reversed SDE, the Kullback-Leibler divergence between $p$ and $p_{\tilde{0}}$ relates to the covariance matrix $\Sigma$ as: $KL(p \parallel \tilde{p}_0) \le \Psi_T + \Psi_{\Delta t} + \Psi_{T,\Delta t}$, with:*

$$\Psi_T = f\left(e^{-4T} \left|\text{Tr}\left((\Sigma - \text{Id})\Sigma\right)\right|\right), \tag{59}$$

$$\Psi_{\Delta t} = f\left(\Delta t \left|\text{Tr}\left(\Sigma^{-1} - \Sigma(\Sigma - \text{Id})^{-1} \log(\Sigma)/2 + (\text{Id} - \Sigma^{-1})/3\right)\right|\right), \tag{60}$$

$$\Psi_{T,\Delta t} = o(\Delta t + e^{-4T}), \qquad \Delta t \to 0, T \to +\infty \tag{61}$$

*where $f(t) = t - \log(1 + t)$ and $d$ is the dimension of $\Sigma$, $\text{Tr}(\Sigma) = d$.*

**Proposition 2.** *For any $\epsilon > 0$, there exists $T, \Delta t \geq 0$ such that:*

$$(1/d)(\Psi_T + \Psi_{\Delta t}) \leq \epsilon, \tag{62}$$

$$T/\Delta t \leq C\epsilon^{-2}\kappa^3, \tag{63}$$

*where $C \geq 0$ is a universal constant, and $\kappa$ is the condition number of $\Sigma$.*

Guth et al. (2022b) provides the proof outline for Theorem 1, based on the following Theorem 2,

**Theorem 2.** *Let $N \in \mathbb{N}$, $\Delta t > 0$, and $T = N\Delta t$. Then, we have that $\bar{x}_t^N \sim \mathcal{N}(\hat{\mu}_N, \Sigma^{\widehat{N}})$ with*

$$\Sigma^{\widehat{N}} = \Sigma + \exp(-4T)\Sigma^{\widehat{T}} + \Delta t\Psi^{\widehat{T}} + (\Delta t)^2 R^{\widehat{T},\Delta t}, \tag{64}$$

$$\hat{\mu}_N = \mu + \exp(-2T)\hat{\mu}_T + \Delta t e^{\widehat{T}} + \frac{(\Delta t)^2}{2}r^{T,\Delta t}, \tag{65}$$

*where $\Sigma^{\widehat{T}}, \Psi^{\widehat{T}}, R^{T,\Delta t} \in \mathbb{R}^{d \times d}$, $\hat{\mu}_T, e^{\widehat{T}}, r^{T,\Delta t} \in \mathbb{R}^d$, and $\|R^{T,\Delta t}\| + \|r^{T,\Delta t}\| \leq R$, not dependent on $T \geq 0$ and $\Delta t > 0$. We have that*

$$\Sigma^{\widehat{T}} = -(\Sigma - \mathrm{Id})(\Sigma\Sigma^{-1})^2, \tag{66}$$

$$\Psi^{\widehat{T}} = \mathrm{Id} - \frac{1}{2}\Sigma^2(\Sigma - \mathrm{Id})^{-1}\log(\Sigma) + \exp(-2T)\Psi^{\widetilde{T}}. \tag{67}$$

*In addition, we have*

$$\hat{\mu}_T = -\Sigma^{-1}T\Sigma\mu, \tag{68}$$

$$e^{\widehat{T}} = \left\{-2\Sigma^{-1} - \frac{1}{4}\Sigma(\Sigma - \mathrm{Id})^{-1}\log(\Sigma)\right\}\mu + \exp(-2T)\widetilde{\mu}_T, \tag{69}$$

*with $\Psi^{\widetilde{T}}, \widetilde{\mu}_T$ bounded and not dependent on $T$.*

**Theorem 3.** *Suppose that $\nabla \log p_t(x)$ is $\varphi^2$ in both $t$ and $x$ such that:*

$$\sup_{x,t}\left\|\nabla^2 \log p_t(x)\right\| \leq K, \qquad \|\partial_t \nabla \log p_t(x)\| \leq Me^{-\alpha t}\|x\| \tag{70}$$

*for some $K, M, \alpha > 0$. Then, $\|p - \tilde{p}_0\|_{TV} \leq \Psi_T + \Psi_{\Delta t} + \Psi_{T,\Delta t}$, where:*

$$\Psi_T = \sqrt{2}e^{-T}\,\mathrm{KL}\left(p \,\|\, \mathcal{N}(0, \mathrm{Id})\right)^{1/2} \tag{71}$$

$$\Psi_{\Delta t} = 6\sqrt{\Delta t}\left[1 + \mathbb{E}_p\left(\|x\|^4\right)^{1/4}\right]\left[1 + K + M\left(1 + 1/2\alpha\right)^{1/2}\right] \tag{72}$$

$$\Psi_{T,\Delta t} = o\left(\sqrt{\Delta t} + e^{-T}\right) \qquad \Delta t \to 0, T \to +\infty \tag{73}$$

Theorem 3 generalizes Theorem 1 to non-Gaussian processes. Please refer to Guth et al. (2022a) for the complete proof.

## A.9 DATASETS

**CelebA-HQ** Karras et al. (2017) dataset is an extension of the original CelebA Liu et al. (2015) dataset. It contains high-quality images of celebrity faces at a higher resolution compared to the original CelebA dataset. This dataset was created to cater to the needs of tasks that require high-resolution facial images. The images in the CelebA-HQ dataset are typically at a resolution of 1024x1024 pixels, providing a significant improvement in image quality compared to the original CelebA dataset, which had lower-resolution images. Similar to the original CelebA dataset, CelebA-HQ comes with a set of facial attribute annotations. These annotations include information about attributes such as gender, age, and presence of accessories like glasses. CelebA-HQ also contains a substantial number of images. While the exact number may vary depending on the specific release, it generally consists of thousands of high-resolution images of celebrity faces. The dataset includes images of a diverse set of celebrities, covering a wide range of genders, ethnicities, and ages.

**Animal Faces-HQ (AFHQ)** dataset, initially introduced in Choi et al. (2020), comprises 15,000 high-resolution images with $512 \times 512$ pixels. This dataset has three distinct domains: cat, dog, and wildlife, each containing 5,000 images. With three domains and a diverse array of breeds ($\geq$ eight) per domain, AFHQ presents a more intricate image-to-image translation challenge. All images are meticulously aligned both vertically and horizontally to position the eyes at the center. The dataset underwent a careful curation process, with low-quality images being manually excluded. In this work, we exclusively utilize the cat and dog categories in all experiments.

## A.10 EVALUATION METRICS

**Kullback-Leibler Divergence** We utilized KL divergence to quantify the similarity between the distribution of LL coefficients and the standard Gaussian distribution $\mathcal{N}_0 = \mathcal{N}(\mathbf{0}, \mathbf{I})$. Suppose the image dataset of interest has $N$ images $\boldsymbol{X}_i \in \mathbb{R}^{L^2}$, $i = 1, 2, \cdots, N$. We could calculate the sample mean and covariance matrix as below:

$$\hat{\boldsymbol{\mu}} = \frac{\sum_{i=1}^N \boldsymbol{X}_i}{N} \tag{74}$$

$$\hat{\Sigma} = \frac{\sum_{i=1}^N \boldsymbol{X}_i \boldsymbol{X}_i^T}{N} - \hat{\boldsymbol{\mu}} \hat{\boldsymbol{\mu}}^T \tag{75}$$

Due to the non-negative data range of image data, we perform pixel-wise normalization to the sample mean and covariance. Denote $\Lambda = (diag(\hat{\Sigma}))^{\frac{1}{2}}$, the normalized sample mean and covariance are

$$\tilde{\boldsymbol{\mu}} = \hat{\boldsymbol{\mu}} - \mathbf{1}^T \hat{\boldsymbol{\mu}} \tag{76}$$

$$\tilde{\Sigma} = \Lambda^{-1} \hat{\Sigma} \Lambda^{-1} \tag{77}$$

By CLT, we know the normalized sample distribution can be approximate by a $L^2$-dimension Gaussian distribution $\mathcal{N}_1 = \mathcal{N}(\tilde{\boldsymbol{\mu}}, \tilde{\Sigma})$. Therefore, the KL divergence between the normalized sample distribution and standard Gaussian is:

$$D_{KL}(\mathcal{N}_0 \parallel \mathcal{N}_1) = \frac{1}{2} \left( \text{tr}(\tilde{\Sigma}^{-1}) - L^2 + \tilde{\boldsymbol{\mu}}^T \tilde{\Sigma}^{-1} \tilde{\boldsymbol{\mu}} + \ln(\det \tilde{\Sigma}) \right) \tag{78}$$

## A.11 ADDITIONAL RESULT

Here we present the sampling results on the CelebA-HQ (5k) and AFHQ-Cat datasets.

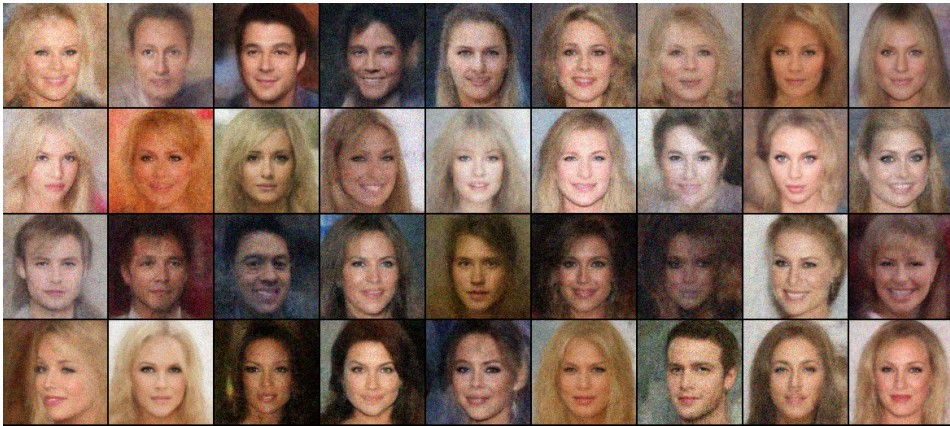

Figure 8: SGM with 4 discretization steps

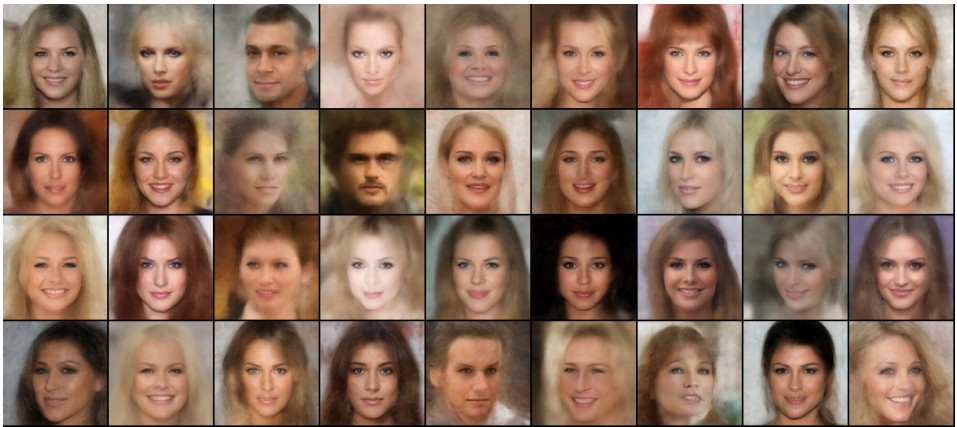

Figure 9: WSGM with 4 discretization steps

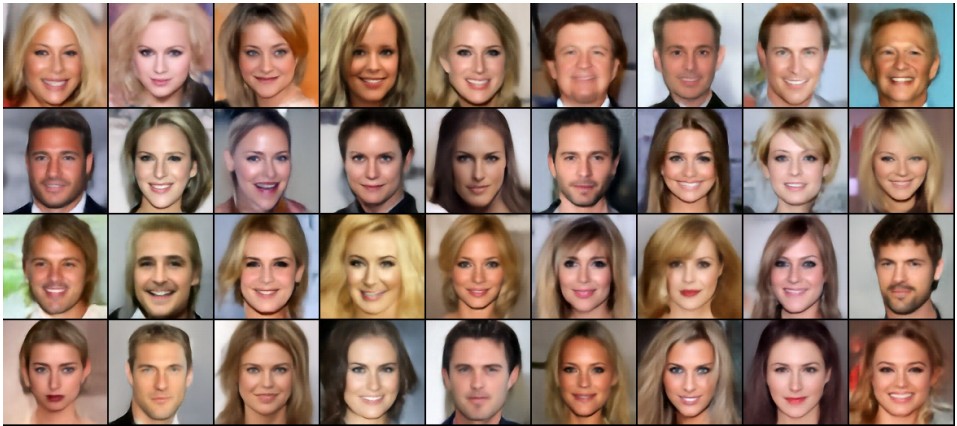

Figure 10: WMGM with 4 discretization steps

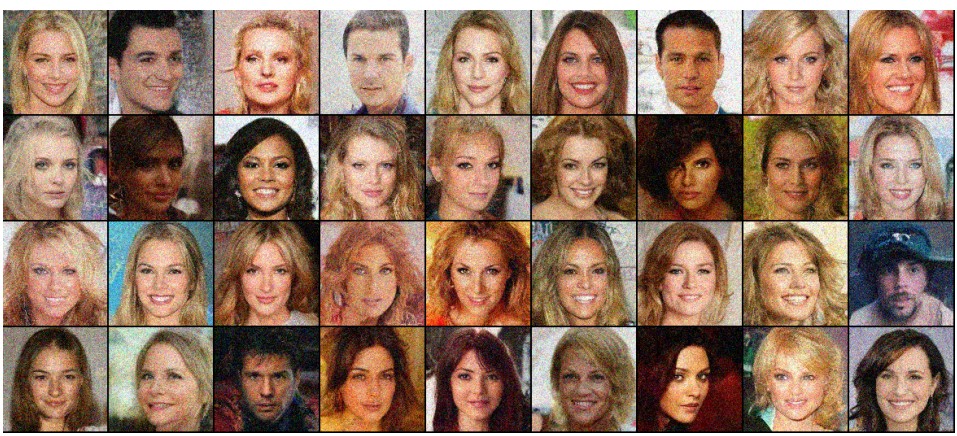

Figure 11: SGM with 16 discretization steps

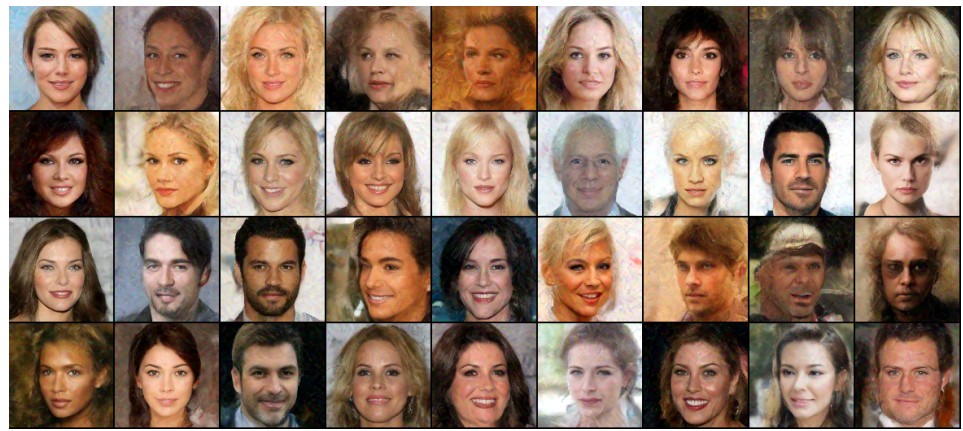

Figure 12: WSGM with 16 discretization steps

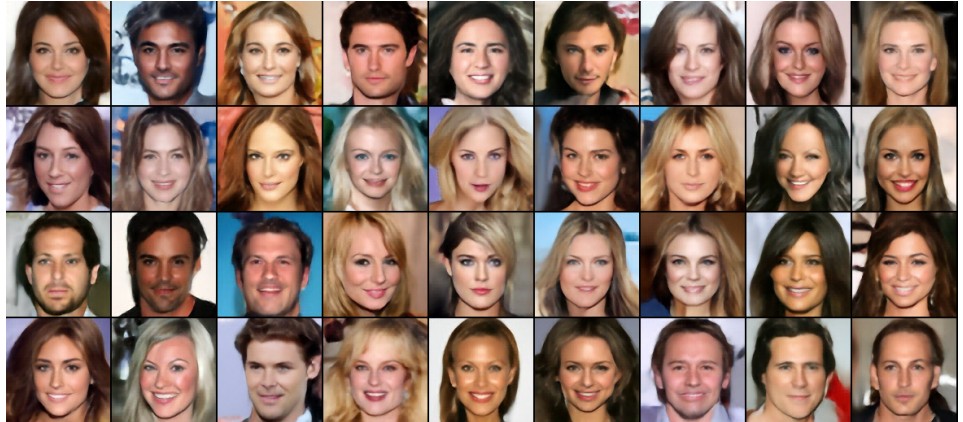

Figure 13: WMGM with 16 discretization steps

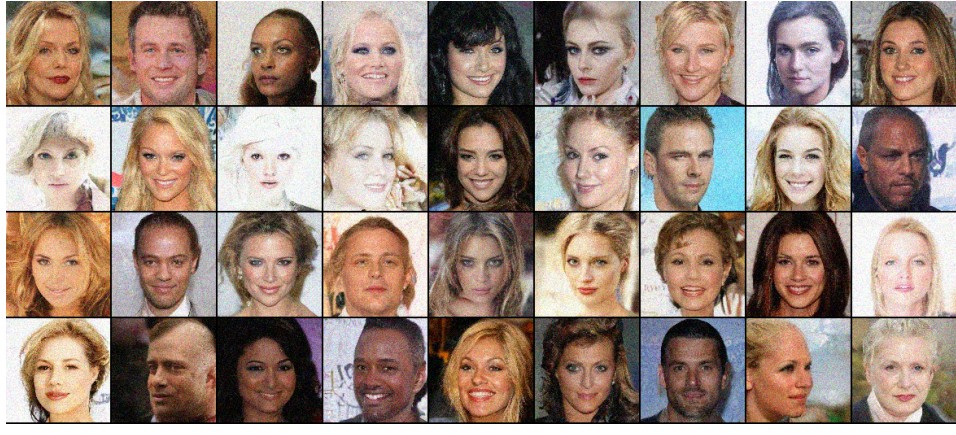

Figure 14: SGM with 64 discretization steps

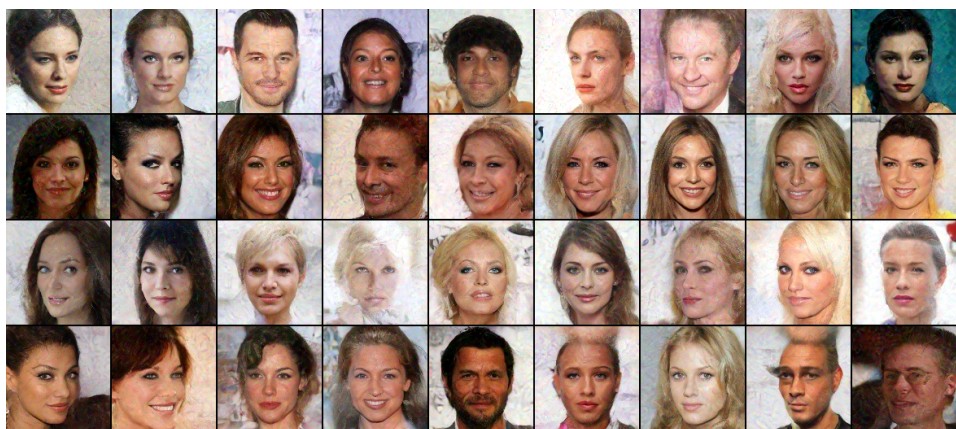

Figure 15: WSGM with 64 discretization steps

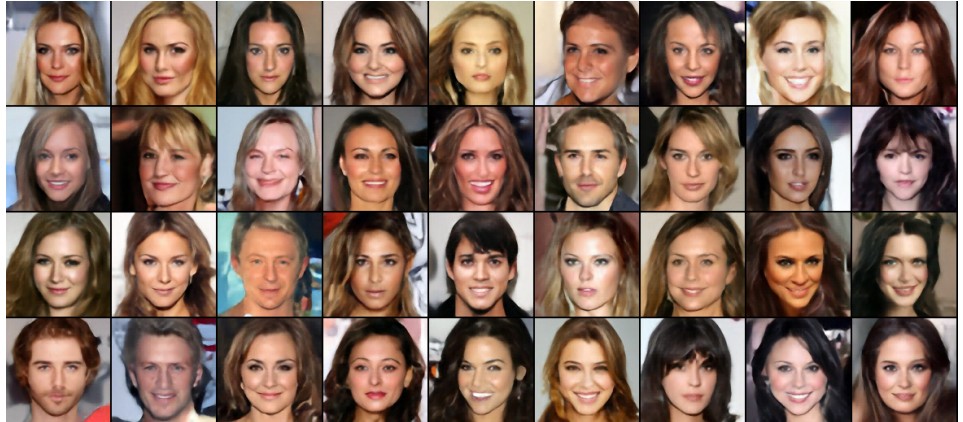

Figure 16: WMGM with 64 discretization steps

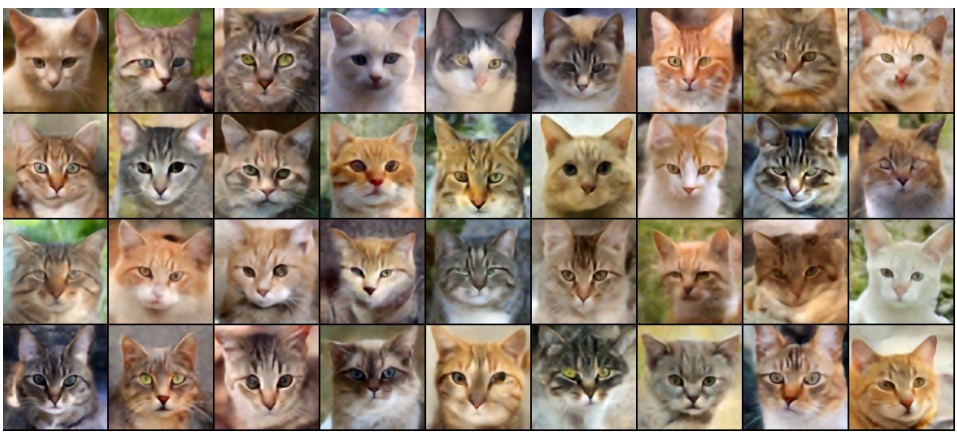

Figure 17: Cat samples with WMGM using 4 discretization steps

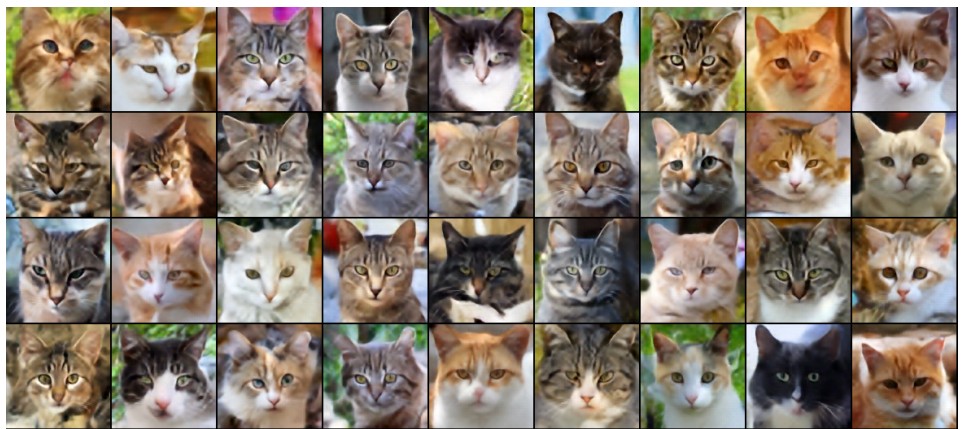

Figure 18: Cat samples with WMGM using 16 discretization steps

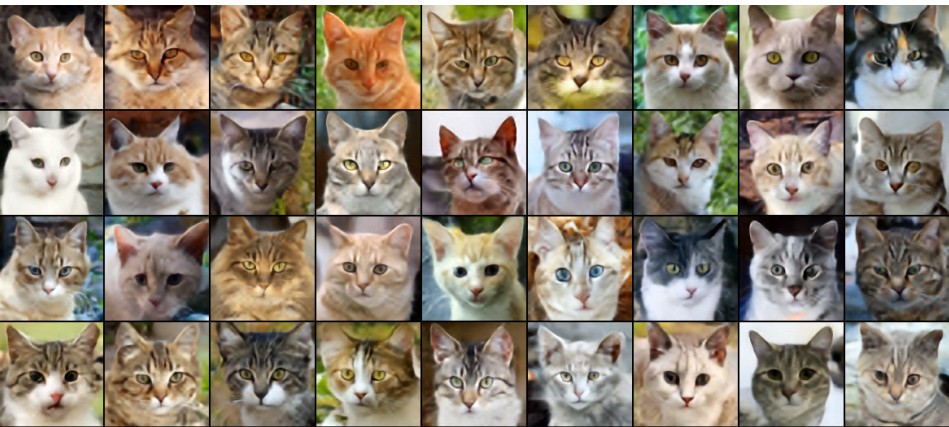

Figure 19: Cat samples with WMGM using 64 discretization steps

