# OpenReview forum: "Multi-Scale Generative Modeling in Wavelet Domain"
_ICLR.cc/2024/Conference — Submitted to ICLR 2024_

### Official Review · Reviewer_UdWx · 2023-10-30

**Soundness:** 2 fair
**Presentation:** 2 fair
**Contribution:** 2 fair
**Rating:** 3
**Confidence:** 4

**Summary:**

This submission deals with generative learning in the wavelet domain. Learning in the wavelet domain is  challenging due to the sparse and correlated nature of coefficients that make it difficult to denoise Gaussian noise for score based generative models (SGM). This submission proposes a multi-scale GANO in the wavelet domain that uses low frequency information to condition learning the high frequency one. The low frequency LL band is learned using a SGM and then the other bands are learned using GANO conditioned on LL band. The LL band is well conditioned, thus learning SGM is easy. Experiments with FFHQ-cat and CelebA datasets show improvements in terms of FID for smaller number of timesteps and smaller architectures.

**Strengths:**

The idea of learning based on only the LL subband for well-conditioned score learning is interesting and innovative

**Weaknesses:**

This work lacks experiments for real scenarios to test generation. FFHQ-cat and CelebA are toy datasets to make a conclusion about the effectiveness of the method. CelebA dataset is known to have very compressible wavelet representation, with a very narrow distribution. It needs more experiments with more realistic scenarios such as imageNet and more ablations to make any conclusions. Even the CIFAR dataset with more classes has not been tested.

The contributions compared with the previous wavelet based SGM method (WSGM) seems not significant. The already have shown acceleration due to wavelet compression.

**Questions:**

For learning high frequency subbands from the LL subband, GANO is used? The motivation behind operator learning, that deals with functional mapping, is not clear here. Why not simply use a Unet and do regression? Or if you want to learn the conditional probability, why not a simple generative superresolution method such as another diffusion? The conditioning of LL bands is very informative to guide any generative method. Ablations are needed to justify the choice of GANO.

---

> ### Author Response · Authors · 2023-11-22
> **Response to Reviewer UdWx**
>
> Thank you for your thorough review of our paper and for recognizing our innovative idea of learning based on only the LL subband with well-conditioned scores. In the following, we offer our detailed responses and clarifications to address the concerns you have mentioned.
>
>  **Lack of datasets:** We are grateful for your suggestions regarding supplementing our dataset. In our experiments, we followed the experimental setup of WSGM [1], utilizing the Celeb-A dataset, and additionally incorporated the FFHQ dataset to enable a more equitable comparison of our experimental results. However, due to the lack of computational resources in our small and poor team, we have not yet completed experiments on ImageNet since it's a really high resolution dataset. We are committed to updating the results in the future revised version. Once our paper is published, we will make our code fully public and we look forward to our model being used on various datasets.
>
> **Compared with WSGM:** In the theoretical realm, the paper WSGM assumes that in the wavelet domain, the high-frequency coefficient distribution under low-frequency conditions is close to Gaussian. Our research challenges this assumption. We have substantiated the sparsity of this conditional distribution in Appendix A.5 (Page 16-17). This finding underpins our rationale for employing distinct models to process high and low-frequency coefficients, which is one of our key contributions. From an experimental perspective, under the same experimental settings, our model demonstrates significant improvements over WSGM. With a parameter count of 89M, our model has only one-fourth the parameters of WSGM, which has 351M. Additionally, our model's sampling speed, at 4175 ms/pic, is 1.78X faster than WSGM's 11097 ms/pic. Moreover, the quality of the images generated by our model still surpasses that of WSGM.
>
> **Why we choose GANO:** We use GANO to handle multi-scale subbands in the wavelet domain. Different from the spatial domain commonly studied by UNet and other super-resolution networks, the multi-scale wavelet domain has less spatial correlation due to its inherent orthogonality. Just like other transformed domains such as the Fourier domain, operator learning has been shown to be more effective and have unique strengths like zero-shot super-resolution ability [2][3].
>
> [1] Guth, Florentin, et al. "Wavelet score-based generative modeling." Advances in Neural Information Processing Systems 35 (2022): 478-491.
>
> [2] Li, Zongyi, et al. "Fourier neural operator for parametric partial differential equations." arXiv preprint arXiv:2010.08895 (2020).
>
> [3]Gupta, Gaurav, Xiongye Xiao, and Paul Bogdan. "Multiwavelet-based operator learning for differential equations." Advances in neural information processing systems 34 (2021): 24048-24062.

---

### Official Review · Reviewer_3QFf · 2023-11-01

**Soundness:** 4 excellent
**Presentation:** 4 excellent
**Contribution:** 4 excellent
**Rating:** 8
**Confidence:** 2

**Summary:**

This mainly theoretical paper attempts to couple wavelet-domain diffusion to its spatial counterpart.
The paper proceeds with an examination of the distribution of high-frequency wavelet coefficients on the CelebA-HQ dataset. In a next step, their non-Gaussian nature is established theoretically. With the non-gaussian nature of the data in mind, the authors propose to use a multi-scale generative adversarial neural operator instead of a diffusion model, which sometimes makes Gaussian assumptions.
Finally, speedups are experimentally observed for the proposed model.

**Strengths:**

- The paper is well written, its research question is a good fit for ICLR.
- Experiments are most likely reproducible. The code is available online.
- To the best of my knowledge, the paper's contributions are novel. Especially the examination of the generative adversarial setup in the wavelet domain. After all, Guth et al. studied only the diffusion case.
- Claims are backed up by extensive material in the supplementary part.

**Weaknesses:**

- It would be nice if the experimental results were statistically significant, that is, if multiple seeds had been tried. However, since the paper is theoretical, I don't think this is an important issue.
- The Gaussian noise assumption is not crucial for working diffusion models [1]. It would have been fair to mention as much.

[1] Cold Diffusion: Inverting Arbitrary Image Transforms Without Noise,  https://arxiv.org/pdf/2208.09392.pdf

**Questions:**

- Why is the proof of section 2.2 in Guth et al. insufficient to establish the duality of spatial and wavelet domain?

---

> ### Author Response · Authors · 2023-11-22
> **Response to Reviewer 3QFf**
>
> Thank you for your meticulous review and valuable insights. We sincerely appreciate your recognition of the novelty of our contribution. In the following, we offer our detailed responses and clarifications to address the concerns you have highlighted.
>
> **1. Lack reference mention.**
>
>
> Thank you for sharing the interesting related work [1]. The work explores the possibility of using deterministic transformations instead of random Gaussian noise in diffusion models. We have added the work in the related work section in the revised version for a broader audience.
>
> **2. Duality proof.**
>
> The section 2.2 in Guth et al. [2] only analyzes the regularity condition of SGM, without explicitly proving the duality between diffusion in spatial and wavelet domains.
>
> [1] Cold Diffusion: Inverting Arbitrary Image Transforms Without Noise, https://arxiv.org/pdf/2208.09392.pdf
>
> [2]. Guth, Florentin, et al. "Wavelet score-based generative modeling." Advances in Neural Information Processing Systems 35 (2022): 478-491.

---

### Official Review · Reviewer_LWzA · 2023-11-06

**Soundness:** 1 poor
**Presentation:** 1 poor
**Contribution:** 1 poor
**Rating:** 1
**Confidence:** 3

**Summary:**

The authors propose a method for image sampling based on wavelet image decompositions. The proposed method generates images using a two step procedure. First, using a score-based model, the method generates a sample of low-frequency wavelet coefficients. As best I can tell, the second step of the sampling procedure involves generating high frequency wavelet coefficients using a conditional GAN.

The authors provide some heuristic arguments that GAN architectures are better suited for sampling high frequency coefficients because the coefficients are sparse and highly non-Gaussian, whereas low-frequency wavelet coefficients tend to be more Gaussian and tend to have a covariance matrix with improved conditioning. Finally, the authors show some experiments in which the proposed method has improved FID and sampling time compared to original SGM (Song et al. 2020) and WSGM (Guth et al. 2022b).

**Strengths:**

The proposed method performs well in the experiment shown in Table 1. The samples generated in Figure 3 are reasonable quality.

**Weaknesses:**

This paper is egregiously vague in many aspects, some of which are: explanation of the proposed method, motivation for the proposed method, and experimental results.

- Explanation of the proposed method. There does not appear to be any description of key details of the GAN architecture. What is the structure of the generator? Figures 1 and 7 are insufficient and more details are required. How large are its layers? How do you implement skip connections and attention gates? Why is it a neural operator? Does it involve in any way a hard-coded wavelet transform? In what sense does the generator sample high-frequency coefficients in 'one shot'? In what sense is the generator 'multi-scale'? What is the intended scale of the coefficients output by the generator, and why does it make sense to minimize terms like $(G(x^k_L, z^k) - x^k_H)^2$ in (17) and (18) if $G$ does not depend on $k$? Please define the parameters lambda, nu, and alpha, and mention how they are chosen in experiments. It does not seem possible to replicate the experiments in this work based on the details provided.

- Motivation for the proposed method. The authors claim repeatedly that low-frequency wavelet coefficients are better conditioned and 'more Gaussian,' and that high-frequency wavelet coefficients are sparse. To support the first claim, the authors show in Figure 5 that KL divergence between a unit gaussian $\mathcal{N}_0$, and a Gaussian with mean and covariance matched to the data $\mathcal{N}_1$, is decreasing as the scale increases. Why does $\text{KL}(\mathcal{N}_0 \mid \mathcal{N}_1)$ have anything to do with well-conditioning of covariance and/or Gaussianity of the data distribution? The discussion in A.4 is vague and extremely non-rigorous. The second claim, about sparsity of the high frequency coefficients, is discussed alongside some supporting citations, but it would be helpful to choose a subset of plots in Figure 6 to show in the body to at least demonstrate these claims empirically. Proposition 1 in A.5 only shows that high average sparsity of $x^k_H$ can imply high average sparsity of $x^k_H$ conditional on $x^k_S$, but I don't understand why this is relevant.

- Experimental results. The comparison to Song et al. 2020 is unfair in light of much followup work on tuning score-based samplers (for example: Score SDE, Song et al. 2021). This method should also be compared to existing work on accelerating diffusion sampling with feed-forward nets, such as Consistency Models (Song et al. 2023) and SBGM in Latent Space (Vahdat et al. 2021). Also, in Figure 4, it's unclear whether the GAN upsampling step of the proposed method is counted as a sampling step. If the comparison is made between FIDs of SGM at 16 steps, WSGM at 16 steps, and 16 steps of WSGM for down-sampled images + GAN upsampling, then it is an unfair comparison in which the proposed method will obviously win. Ideally, the authors should compare to other methods that combine score-based sampling and feedforward nets, and they should demonstrate that wavelet-based architectures can augment this approach.

**Questions:**

See weaknesses section

---

> ### Author Response · Authors · 2023-11-22
> **Response to Reviewer LWzA**
>
> Thank you for the time and effort you have invested in reviewing our paper. We realize that you may have spent considerable time trying to understand our work. However, there seems to be a significant gap stemming from the fact that we are not on the same page regarding certain aspects of our research. To remedy this, we wish to first elucidate the motivation and background behind our work, which originates from analyzing the noise-adding process in the frequency domain (please refer to Figure 2 (Page 3) and Appendix A.2 (Page 13)). This analysis revealed a significant disparity in how high-frequency and low-frequency subbands respond to noise. In the frequency domain, the noise impacts high- and low-frequency components differently, resulting in varying degradation rates. This uneven degradation provided the inspiration to treat signals of different frequencies separately, prompting us to develop distinct strategies for their optimization. We have also written such inspiration in our paper (Page 2) as follows:
>
> "*To make the discussion more concrete, let us consider the noise-adding process in the frequency domain (i.e., wavelet domain), where noise contains a uniform power spectrum in each frequency band. However, in the wavelet domain, the high-frequency coefficients of the images are sparse and contain minimal energy, while the low-frequency coefficients encapsulate most of the energy. Given the disparity between image and noise power spectra, low-frequency components, which hold the majority of the energy, receive the same magnitude of noise during the noise addition process, and high-frequency coefficients, despite being sparse, obtain a relatively larger amount of noise. This dynamic offers inspiration for analyzing diffusion in the frequency domain, incorporating corresponding generative strategies tailored for each frequency sub-domain (namely, the high and low-frequency domains).*"
>
> Furthermore, we have elucidated these observations with relevant theories provided in [1], specifically by addressing the ill-conditioned problems arising from the power-law decay inherent in natural images. Related works [1,2] also suggests through theoretical and quantitative analysis of the score $\nabla \log p_{t}$ that closer adherence to a Gaussian distribution results in higher score regularity, leading to fewer required sampling steps. We have also highlight this in our paper in page 4 as:
>
> "*Here, the minimum number of time steps $N=\frac{T}{\Delta t}$ is limited by the Lipschitz regularity of the score $ \nabla \log p_{t} $, as detailed in Theorem 1 [1] and Theorem 2 [2].* "
>
> Therefore, we discuss the distribution characteristics of high and low-frequency coefficients in the wavelet domain, and further provide the duality proof of diffusion in the wavelet domain.
>
> We hope that the motivation and background we have provided will help bring us to the same stage of understanding. To further facilitate this alignment, we provide the point-wise response addressing each specific concern you raised below:
>
> [1]. Guth, Florentin, et al. "Wavelet score-based generative modeling." Advances in Neural Information Processing Systems 35 (2022): 478-491.
>
> [2]. De Bortoli, Valentin, et al. "Diffusion Schrödinger bridge with applications to score-based generative modeling." Advances in Neural Information Processing Systems 34 (2021): 17695-17709.

---

> ### Author Response · Authors · 2023-11-22
> **Response to the concerns about "Explanation of the proposed method."**
>
> **1.Explanation of the proposed method.**
>
> - ` There does not appear to be any description of key details of the GAN architecture. What is the structure of the generator? Figures 1 and 7 are insufficient and more details are required. How large are its layers? How are skip connections and attention gates implemented?`
>
> **What is the structure of the generator?**
>
> The generator's architecture is fundamentally based on the UNet structure and additionally integrates attention modules in the upsampling stages. Upon entering the generator, the data undergoes a sequence of five downsampling layers, followed by an equal number of upsampling layers. Post each upsampling phase, the data is processed through an attention module. After these sequential operations, the data is subject to two convolutional layers and corresponding activation functions, culminating in the output of the required values.
>
> **How large are its layers?**
>
> The Attention-GAN model comprises approximately 36 million parameters and encompasses a depth of 106 layers. The model's architecture initiates with the width corresponding to the input channel size, progresses through a downsampling phase reaching a maximum width of 1024, and subsequently transitions to the width of the output channel following the upsampling and attention processes. The width of the intermediate layers is adjustable through parameter modification.
>
>
> **How are skip connections and attention gates implemented?**
>
> In this model, the skip connections are uniquely characterized by incorporating corresponding downsampling information during the upsampling stages. The distinctive aspect is that the data, post-processing by the upsampling layers, are utilized as input for the attention gates. These are then integrated into the attention block along with the corresponding downsampling information. The implementation of the attention gate is executed as follows:
>
> + The transformations **self.W_g** and **self.W_x** are first applied to the gating signal **g** and the local signal **x**, respectively. This process yields two intermediate feature maps.
> + These feature maps are subsequently combined and channeled through a ReLU activation function and a 2D Convolutional layer, resulting in the creation of the attention map.
> + Lastly, the attention map undergoes an element-wise multiplication with the original local signal **x**. This operation generates the output of the attention block, which selectively accentuates certain features of **x** based on the gating signal **g**.
>
> The comprehensive details about the GAN architecture were provided in the supplementary materials, including the structure of the generator. The entire source code can be found in Appendix A.1 (Page 13). We have also added the details in the revised vision (Please refer to Appendix A.6 Attention-GAN Architecture Description, Page 18).

---

> ### Author Response · Authors · 2023-11-22
> **Continue**
>
> - `Why is it referred to as a neural operator?`
>
> In our research, we employ a methodology that aligns closely with the concept of neural operators [3-5]. Specifically, we project images into the wavelet domain and then focus on learning the mappings between wavelet coefficients of wavelet functions. This process essentially involves learning function-to-function mappings within the wavelet space, a hallmark of neural operator frameworks [3,4]. By transforming images into wavelet coefficients and subsequently learning their interrelationships, our model effectively operates within the function space of these coefficients. Given the nature of this task — mapping sets of wavelet coefficients to others — describing our approach as a form of neural operator learning is more appropriate.
>
> - `Does it involve any hard-coded wavelet transform?`
>
> The wavelet transform method mentioned in this paper is hard-coded. In our model, we use a 2D Discrete Haar Wavelet Transform [6] (please refer to Appendix A.7 in page 19) to extract frequency domain information. The wavelet transform method applies a pair of filters (low-pass and high-pass) both horizontally and vertically to the image, and then downsamples the result. This process generates four sets of wavelet coefficients, representing the image's low-frequency approximation and high-frequency detail information.
>
> - `In what sense does the generator sample high-frequency coefficients in 'one shot'?`
>
> Here ‘one shot’ is claimed in contrast to most diffusion models that require iterative inferences. The generator samples high-frequency components based on the corresponding low-frequency components at each scale in one single forward inference. Consequently, this “one shot” capability greatly improves the sampling speed of our method.
>
> - `In what sense is the generator 'multi-scale'?`
>
> The generator in our method is multi-scale as it is applied to generating high frequency components at various wavelet scales, without the need to train or fine tune its parameters separately for each scale.
>
> - `What is the intended scale of the coefficients output by the generator, and why does it make sense to minimize terms like` $ \left ( G\left ( x_{L}^{k} ,z^{k} \right ) - x_{H}^{k}\right ) ^{2} $` in equations (17) and (18) if` $ G $` does not depend on `$k$?
>
> The spatial dimension of the output coefficients by the generator will be the same as the input coefficients and the channel of the output coefficients will be tripled. For example, suppose the input low-frequency coefficients $ x_L^k $ is of dimension $H·W·C$, the output coefficients, representing the corresponding high-frequency coefficients is of dimension $H·W·3C$. In Eq. (17) and (18) , G and D are generator and discriminator with trainable/optimizable parameters, respectively. As a common practice in most GAN papers [7], the trainable parameters are omitted in the expression of loss terms.
>
> [3]. Li, Zongyi, et al. "Fourier neural operator for parametric partial differential equations." arXiv preprint arXiv:2010.08895 (2020).
>
> [4]. Gupta, Gaurav, Xiongye Xiao, and Paul Bogdan. "Multiwavelet-based operator learning for differential equations." Advances in neural information processing systems 34 (2021): 24048-24062.
>
> [5]. Kovachki, Nikola, et al. "Neural operator: Learning maps between function spaces." arXiv preprint arXiv:2108.08481 (2021).
>
> [6]. Zhang, Dengsheng, and Dengsheng Zhang. "Wavelet transform." Fundamentals of image data mining: Analysis, Features, Classification and Retrieval (2019): 35-44.
>
> [7]. Goodfellow, Ian, et al. "Generative adversarial networks." Communications of the ACM 63.11 (2020): 139-144.

---

> ### Author Response · Authors · 2023-11-22
> **Response to the concerns about "Motivation for the proposed method."**
>
> **2.Motivation for the proposed method.**
>
> Firstly, we would like to point out a potential misunderstanding here and emphasize the $KL$ divergence and Fig. 5 (Page 15) experimentally support our statements in Sec 3.2.2 (Page 5) that wavelet decomposition has the whitening effect on low-frequency coefficients. Moreover, we would also like to point out that $N_1$ is the Gaussian of normalized samples drawn from image datasets like CelebA-HQ. The normalization process is detailed in Appendix 10 (Page 23), in order to remove the effects of average pixel intensity and image dynamic range. As shown by Equation (78) (Page 23), the $KL$ divergence measures the similarity between the standard Gaussian $N_0$ and the normalized sample distribution $N_1$. Thus, a higher $KL$ divergence indicates higher non-Gaussianity of the sample distribution, and vice versa. We plotted the $KL$ divergence to support the claim of the whitening effect of multi-scale wavelet transform. As shown in Fig. 5, the LL subband at a coarser scale will have better Gaussianity.
>
> Secondly, we would like to emphasize that Proposition 1 (Page 16) in Appendix A.5 (Page 16, 17) justifies the choice of GANO. In Eq. 13 (Page 6), we factorize the sampling process from true distribution p in the multi-scale wavelet domain, and the term $p(x_H^k|x_L^k)$ is highly non-Gaussian due to the property of high-frequency coefficients. Therefore we chose using a multi-scale GANO instead of other generative learning models such as diffusion models based on the non-Gaussianity of high-frequency coefficients. Combining the fact that $x_H^k$ is sparse, as shown in Fig. 6 (Page 17) and Proposition 1, one can easily draw the conclusion that $x_H^k|x_L^k$ has a high expected sparsity, and therefore supporting its high non-Gaussianity.

---

> > ### Author Response · Authors · 2023-11-22
> > **Response to the concerns about "Experimental results."**
> >
> > **3.Experimental results.**
> >
> > We would like to express our gratitude to the reviewer for bringing attention to the influential literature in the field. It is crucial to clarify that our approach fundamentally diverges from methods that expedite diffusion sampling through feed-forward networks. Instead of training a network to approximate the reverse diffusion process with larger temporal intervals, as most related works do, our methodology involves the refactorization of the image generation process in the multi-scale wavelet domain. Specifically, we have developed the SGM and GANO to handle the coarsest level of low-frequency and high-frequency coefficients, respectively. We have also undertaken a comprehensive theoretical analysis of coefficient distributions, providing a rationale and motivation for including GANO. It is essential to emphasize that GANO is an intrinsic and indispensable component of our approach for generating non-Gaussian high-frequency coefficients, rather than just a substitute for diffusion models.
> >
> > Regarding the concern about "unfair comparison" you raised, our ‘sampling steps’ refers to the total diffusion steps calculated. For instance, in the case of WSGM, which consists of three SGM models, 16 sampling steps represent the total number of steps across these three SGM models. Similarly, we set the diffusion steps in our model to 16, ensuring a relatively fair comparison. For not including the sampling steps of the GANO model. The reason is two-fold: Firstly, the steps in a GANO model cannot be directly compared with those in a diffusion model due to fundamental differences in their theory and model structure. Secondly, compared to diffusion models, GANOs typically require less time, fewer resources, and have lower model complexity. In the Table 1 of our paper, our model's parameters amount to 75M (Diffusion model) + 14M (GANO), with sampling times of 3823ms (Diffusion model) + 352ms (GANO). Both in terms of parameter count and sampling time, our model is significantly more efficient than the other two models under consideration. Therefore, our method of comparison is relatively rigorous and aligns with the goal of our experiment: fast sampling.

---

### Meta-Review · Area_Chair_xeih · 2023-12-13

**Metareview:**

This work considers generative modeling in the wavelet domain, where the low frequency LL band is modeled using a SGM and then the other high frequency bands are modeled using a multi-scale conditional GAN framework conditioned on the LL band. Reviewers find this idea is novel and interesting, and the preliminary evaluation is promising. Major concerns include: 1. the evaluation is far from sufficient to demonstrate its effectivenss in this fast growing field (e.g. to compare with the latent diffusion models where the latent space is in spirit similar to the wavelet domain, and thus it is interesting to compare their advantages for generative modeling); 2. several key arguments need further justifications. AC agrees this work needs further improvement, especially on evaluation.

**Justification For Why Not Higher Score:**

1. the evaluation is far from sufficient to demonstrate its effectivenss in this fast growing field (e.g. to compare with the latent diffusion models where the latent space is in spirit similar to the wavelet domain, and thus it is interesting to compare their advantages for generative modeling); 2. several key arguments need further justifications.

**Justification For Why Not Lower Score:**

N/A

---

### Decision · Program_Chairs · 2024-01-16

Reject